# Statistical and Computational Trade-off in Multi-Agent Multi-Armed Bandits

**Filippo Vannella**
KTH and Ericsson Research
Stockholm, Sweden
vannella@kth.se

**Alexandre Protiuere**
KTH
Stockholm, Sweden
alepro@kth.se

**Jaeseong Jeong**
Ericsson Research
Stockholm, Sweden
jaeseong.jeong@ericsson.com

## Abstract

We study the problem of regret minimization in Multi-Agent Multi-Armed Bandits (MAMABs) where the rewards are defined through a factor graph. We derive an instance-specific regret lower bound and characterize the minimal expected number of times each global action should be explored. This bound and the corresponding optimal exploration process are obtained by solving a combinatorial optimization problem whose set of variables and constraints exponentially grow with the number of agents, and cannot be exploited in the design of efficient algorithms. Inspired by Mean Field approximation techniques used in graphical models, we provide simple upper bounds of the regret lower bound. The corresponding optimization problems have a reduced number of variables and constraints. By tuning the latter, we may explore the trade-off between the achievable regret and the complexity of computing the corresponding exploration process. We devise Efficient Sampling for MAMAB (ESM), an algorithm whose regret asymptotically matches the approximated lower bounds. The regret and computational complexity of ESM are assessed numerically, using both synthetic and real-world experiments in radio communications networks.

## 1   Introduction

The stochastic Multi-Agent Multi-Armed Bandits (MAMABs) [35, 2, 3] is a combinatorial sequential decision-making problem that generalizes the classical stochastic MAB problem by assuming that $(i)$ a *global action* is defined by actions individually selected by a set of agents, and $(ii)$ the *reward* function is defined through a factor graph, which defines inter-dependencies between agents. This reward structure arises naturally in applications where agents interact in a graph with the need to coordinate towards a common goal. MAMABs can model a wide range of real-world problems, from wind farm control [2, 37] to radio communication networks parameters optimization (see Fig. 1).

Despite the wide spectrum of their potential applications, MAMABs are extremely hard to solve, even when the reward function is known. The main challenge stems from the combinatorial structure of the action set (there are $K^N$ possible global actions, where $N$ is the number of agents and $K$ is the number of actions per agent). This issue is exacerbated in the learning setting where the reward function has to be inferred. In this work, we study the regret minimization problem in MAMABs, and more specifically, the trade-off between statistical efficiency (the learner aims at achieving low regret), and computational efficiency (she will typically have to solve combinatorial optimization problems over the set of possible global actions while learning).

37th Conference on Neural Information Processing Systems (NeurIPS 2023).

**Contributions.** We present statistically and computationally efficient algorithms for MAMABs. Our algorithms enjoy (in the worst case) regret guarantees scaling as $\rho K^d \log(T)$, where $K$ is the number of actions per agent, $\rho$ and $d$ are the number of factors and the maximal degree of the graph defining the reward function. This scaling illustrates the gains one may achieve by exploiting the factor graph structure: without leveraging it, the regret would scale as $K^N \log(T)$. Our algorithms have controllable computational complexity and can be applied in large-scale MAMABs. More precisely, our contributions are as follows.

*1) Regret lower bound.* We derive a regret lower bound satisfied by any algorithm. The bound is defined through a convex program (the *lower bound problem*), whose solution provides an optimal exploration strategy. Unfortunately, because of the factored reward structure, this optimization problem contains an exponential number of variables and constraints, and is hard to use in practice.

*2) Approximations of the lower bound problem.* We devise approximations of the lower bound problem by combining variable and constraint reduction techniques inspired by methods in the probabilistic graphical model literature [39, 20]. To reduce the number of variables, we propose $(i)$ *locally tree-like* approximation, a tight relaxation for MAMAB instances described by acyclic factor graphs, and $(ii)$ *Mean Field* (MF) approximation for general graphs. The MF approximation yields an upper bound of the regret lower bound, scaling as $\rho K^d \log(T)$ (where $T$ is the time horizon).

Both approximations yield lower bound problems with a polynomial number of variables and exponential number of constraints (in $N$). To reduce the number of constraints, we propose a technique that leverages an ordering of the $m$ smallest gaps and a Factored Constraint Reduction (FCR) method to represent the exponentially many constraints in a compact manner. The corresponding optimization problems have a reduced number of variables and constraints. By tuning the latter, we may explore the trade-off between the achievable regret and the complexity of computing the corresponding exploration process.

*3) The ESM algorithm.* Based on this approximation, we devise Efficient Sampling for MAMABs (ESM), an algorithm whose regret provably matches our approximated regret lower bound. The algorithm trades off statistical and computational complexity by performing exploration as prescribed by the solution of the approximated lower bound problem. We test the performance of ESM numerically on both synthetic experiments and learn to coordinate the antenna tilts in a radio communication network. In both sets of experiments, ESM can solve problems with a large number of global actions in a statistical and computationally efficient manner.

## 2 Related Work

Our work belongs to the framework of structured regret minimization in MABs, which encompasses a large variety of reward structures such as linear [21], unimodal [9], Lipschitz [11], etc. For general structured bandits, [8] propose Optimal Sampling for Structured Bandits (OSSB), a statistically optimal algorithm, i.e., matching the regret lower bound. The algorithm is computationally inefficient when applied to the MAMABs combinatorial structure. Our algorithm is inspired by OSSB, but relies on approximated lower bound problems to trade-off statistical and computational complexity.

A few studies investigate MAMABs with the same factored reward structure as ours [35, 2, 37]. These works focus on devising algorithms with regret guarantees using methods based on, e.g., Upper Confidence Bound (UCB) [35, 2] or Thompson Sampling (TS) [37]. For example, Stranders et al. [35] propose HEIST, an UCB-type algorithm whose asymptotic regret scales as $O(K^N \Delta_{\max}/\Delta_{\min} \log(T))$, where $\Delta_{\min}$ and $\Delta_{\max}$ are the minimal and maximal gaps, respectively. The MAUCE algorithm from Bargiacchi et al., [2] improves over [35] yielding asymptotic regret $O(\rho^2 K^d \Delta_{\max}^2/\Delta_{\min}^2 \log(T))$. Our worst approximation improves of a factor $\Delta_{\max}$ w.r.t. this bound, a quantity that typically scales with $\rho K^d$ (see App. M).

There is a large body of work [24, 10, 12, 13, 38] investigating regret minimization in the (linear) *combinatorial semi-bandit feedback* setting. Although our model can be interpreted as a particular instance of this setting (see App. E for details), the MAMAB combinatorial structure has never been explicitly considered in this context. The closest related work is [12], in which the authors study a regret lower bound problem with an exponentially large number of variables and constraints. They leverage [12, Assumption 6] to compactly represent the lower bound optimization problem and propose a gradient-based procedure to solve it in polynomial time. Unfortunately, for MAMABs, the

above-mentioned assumption only holds for rewards described by acyclic factor graphs (see App. M). We propose computationally efficient approximations valid for any factor graph while retaining statistical tightness in the case of acyclic factor graphs.

## 3 Problem Setting

We consider the generic MAMAB model with factored structure introduced in [2]. The model is defined by the tuple $\langle \mathcal{S}, \mathcal{A}, r \rangle$, where:

1. $\mathcal{S} = [N] \triangleq \{1, \ldots, N\}$ is a set of $N$ agents;

2. $\mathcal{A} = \times_{i \in [N]} \mathcal{A}_i$ is a set of global actions, which is the Cartesian product over $i$ of the set $\mathcal{A}_i$ of actions available to the agent $i$. We assume w.l.o.g. that $|\mathcal{A}_i| = K$, for all $i \in [N]$, and define $A \triangleq |\mathcal{A}| = K^N$;

3. $r$ is the reward function mapping the global action to the collected reward.

**Rewards and their factor-graph representation.** We model the collected rewards. There are $\rho$ possibly overlapping groups of agents $(\mathcal{S}_e)_{e \in [\rho]}$, with $\mathcal{S}_e \subseteq \mathcal{S}$ and $|\mathcal{S}_e| = N_e$. The local reward generated by group $e$ depends on group actions $a_e \triangleq (a_i)_{i \in \mathcal{S}_e} \in \mathcal{A}_e \triangleq \times_{i \in \mathcal{S}_e} \mathcal{A}_i$ only. More precisely, each time $a_e$ is selected, the collected local rewards are i.i.d. copies of a random variable $r_e(a_e) \sim \mathcal{N}(\theta_e(a_e), 1/2)$. Rewards collected in various groups are independent. The global reward for action $a$ is then $r(a) = \sum_{e \in [\rho]} r_e(a_e)$, a random variable with expectation $\theta(a) = \sum_{e \in [\rho]} \theta_e(a_e)$. The number of possible group actions in group $e$ is $A_e \triangleq |\mathcal{A}_e| = K^{N_e}$, and we define $\tilde{A} \triangleq \sum_{e \in [\rho]} A_e$. The reward function can be represented using a factor graph [39]. Factor graphs are bipartite graphs with two types of node: $N$ *action nodes*, one for each agent, and $\rho$ *factor nodes*, one for each group. An edge between a factor $r_e$ and an agent $i$ exists if the action $a_i$ selected by the agent $i$ is an input of $r_e$: $i \in \mathcal{S}_e$. Fig. 1 shows an example of a factor graph modeling interference in a radio communication network.

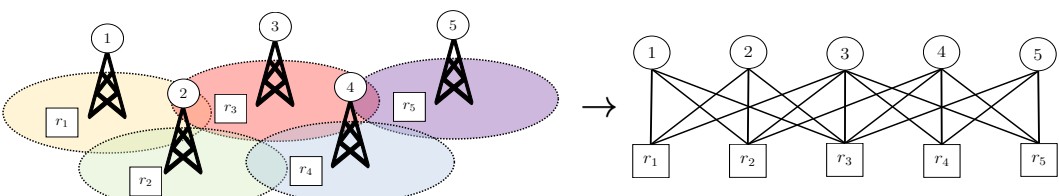

Figure 1: Factor graph in a radio communication network. An agent (represented by a circle) corresponds to a base-station (BS) whose transmissions cover a cell. The possible actions at a BS may correspond to different transmission power levels and antenna tilts (the physical angle of the antennas). The local rewards correspond to the throughput (in bit/s) achieved in a given cell, and hence each cell is associated with a factor (represented by a square). The throughput in a given cell depends on the action of the corresponding BS but also on those of neighboring BSs through interference. In the factor graph, each BS or agent has hence an edge to factors or cells it interferes.

**Sequential decision process.** The decision maker sequentially selects global actions based on the history of previous observations and receives a set of samples of the local rewards associated to the various groups. Specifically, in each round $t \geq 1$, the decision maker selects a global action $a_t = (a_{t,1}, \ldots, a_{t,N})$ and observes the local rewards $r_t = (r_{t,1}, \ldots, r_{t,\rho})$ from each group. The global action $a_{t+1}$ is selected based on the history of observations $\mathcal{H}_t = (a_s, r_s)_{s \in [t]}$. This type of interaction is known as *semi-bandit* feedback.

**Regret minimization.** The goal is to devise an algorithm $\pi = (a_t)_{t \geq 1}$, i.e., a sequence of global actions $a_t \in \mathcal{A}$ selected in each round $t \geq 1$, that minimizes the *regret* up to time $T \geq 1$, defined as

$$R^\pi(T) = \mathbb{E}\left[\sum_{t=1}^{T} \theta(a_\theta^\star) - \theta(a_t)\right],$$

where $a_\theta^\star \in \arg\max_{a \in \mathcal{A}} \theta(a)$ denotes the *best global action*. Throughout the paper, we assume that $a_\theta^\star$ is unique and we use $a^\star$ and $a_\theta^\star$ interchangeably. We define the gap of a sub-optimal global action $a$ by $\Delta(a) = \theta(a^\star) - \theta(a)$.

# 4 Regret Lower Bound

To derive instance-specific regret lower bounds, we restrict our attention to the class of *uniformly good* algorithms: An algorithm $\pi$ is uniformly good if for any $\theta$, $\forall \alpha > 0$, we have that $R^\pi(T) = o(T^\alpha)$.

**Theorem 4.1.** *The regret of any uniformly good algorithm satisfies for any $\theta$,* $\liminf_{T \to \infty} \frac{R^\pi(T)}{\log(T)} \geq C_\theta^\star$, *where $C_\theta^\star$ is the value of the following convex optimization problem*

$$\min_{v \in \mathbb{R}_{\geq 0}^A} \sum_{a \in \mathcal{A}} v_a \Delta(a) \quad \text{s.t.} \quad \sum_{e \in [\rho]: a_e \neq a_e^\star} \left( \sum_{b \in \mathcal{A} \setminus \{a_\theta^\star\}: b_e = a_e} v_b \right)^{-1} \leq \Delta(a)^2, \quad \forall a \in \mathcal{A}. \quad (1)$$

The proof of this result leverages classical change-of-measure arguments [25] (see App. A.1 for details). If $v^\star$ denotes the solution of the lower bound optimization problem, then for $a \neq a_\theta^\star$, $v_a^\star \log(T)$ can be interpreted as the asymptotic expected number of times the sub-optimal action $a$ is explored under a uniformly good algorithm minimizing regret. We conclude this section by reformulating (1) using *group variables* $\tilde{v}$. Introduce the *marginal cone*:

$$\tilde{\mathcal{V}} = \left\{ \tilde{v} \in \mathbb{R}_{\geq 0}^{\tilde{A}} : \exists v \in \mathbb{R}_{\geq 0}^A, \forall e \in [\rho], a_e \in \mathcal{A}_e, \tilde{v}_{e,a_e} = \sum_{b \in \mathcal{A} \setminus \{a_\theta^\star\}: b_e = a_e} v_b, \right\}.$$

The set $\tilde{\mathcal{V}}$ contains group variables $\tilde{v} = (\tilde{v}_e)_{e \in [\rho]}$ where $\tilde{v}_e = (\tilde{v}_{e,a_e})_{a_e \in \mathcal{A}_e}$.

**Lemma 4.2.** *For any $\theta$, $C_\theta^\star$ is the value of the following convex optimization problem*

$$\min_{\tilde{v} \in \tilde{\mathcal{V}}} \sum_{e \in [\rho], a_e \in \mathcal{A}_e} \tilde{v}_{e,a_e} (\theta_e(a_e^\star) - \theta_e(a_e)) \quad \text{s.t.} \quad \sum_{e \in [\rho]: a_e \neq a_e^\star} \tilde{v}_{e,a_e}^{-1} \leq \Delta(a)^2, \quad \forall a \in \mathcal{A}. \quad (2)$$

Again, if the solution of (2) is $\tilde{v}^\star$, then for any $e \in [\rho]$ and $a_e \in \mathcal{A}_e$, $\tilde{v}_{e,a_e}^\star \log(T)$ can be interpreted as the asymptotic expected number of times the group action $a_e$ is selected under an optimal algorithm when it explores, i.e., when the global action $a \neq a_\theta^\star$.

# 5 Lower Bound Approximations

As suggested above, if we are able to solve (1) and hence obtain $v^\star$, the latter specifies the optimal exploration process. From there, we could devise an algorithm with minimal regret [8]. Unfortunately, solving (1) is an extremely hard task, even for relatively small problems. Indeed, the problem has $K^N$ variables and $K^N$ constraints, and using general-purpose solvers, e.g., based on the interior-point method, would require $\text{poly}(K^N) \log(1/\varepsilon)$ floating-point operations [33]. To circumvent this difficulty, we present approximations of the lower bound problem with a reduced number of variables and constraints. We will then leverage these approximations to design efficient algorithms.

## 5.1 Variable reduction

To reduce the number of variables, we apply approximation techniques inspired by methods in the probabilistic graphical model literature [39]. In Sec. 5.1.1, we first propose a *locally tree-like* reduction, yielding an optimization problem whose value $C_\theta^L$ exactly matches the true lower bound $C_\theta^\star$ for MAMABs with acyclic factor graphs (see App. J for a formal definition and examples). For graphs containing cycles however, we have $C_\theta^L < C_\theta^\star$, and hence for those graphs, it is impossible to devise an algorithm based on this reduction (such an algorithm would lead to a regret $C_\theta^L \log(T)$, which contradicts the lower bound).

Instead, for general graphs, we propose in Sec. 5.1.2 the $\psi$-*mean-field* reduction, an approximation based on a local decomposition inspired by Mean Field (MF) methods [39]. The $\psi$-mean field reduction leads to an optimization problem whose value $C_\theta^{\text{MF}}$ provably upper bounds $C_\theta^\star$, and hence that can be used to devise an algorithm with regret approaching $C_\theta^{\text{MF}} \log(T)$.

### 5.1.1 Locally tree-like reduction

This reduction imposes local consistency constraints between group variables $\tilde{v}$ and local variables *local variables* $w = (w_i)_{i \in [N]}$, where $w_i = (w_{i,a_i})_{a_i \in \mathcal{A}_i} \in \mathbb{R}_{\geq 0}^K$. Define the local cone as:

$$\tilde{\mathcal{V}}_L = \left\{ \tilde{v} \in \mathbb{R}^{\tilde{A}} : \exists w \in \mathbb{R}_{\geq 0}^{KN} : \forall e \in [\rho], \forall i \in \mathcal{S}_e, \forall a_i \in \mathcal{A}_i, w_{i,a_i} = \sum_{b_e \in \mathcal{A}_e : b_e \sim a_i} \tilde{v}_{b_e} \right\},$$

where the notation $a_e \sim a_i$ means that the $i^{\text{th}}$ element of $a_e$ equals $a_i$. The locally tree-like approximation, presented in the next lemma, is obtained by replacing $\tilde{\mathcal{V}}$ by $\tilde{\mathcal{V}}_L$ in (2).

**Lemma 5.1.** *For any $\theta$ with rewards described by an acyclic factor graph, we have that $C_\theta^\star = C_\theta^L$, where $C_\theta^L$ is the value of the following convex optimization problem:*

$$\min_{\tilde{v} \in \tilde{\mathcal{V}}_L} \sum_{e \in [\rho], a_e \in \mathcal{A}_e} \tilde{v}_{e,a_e}(\theta_e(a_e^\star) - \theta_e(a_e)) \quad \text{s.t.} \quad \sum_{e \in [\rho]: a_e \neq a_e^\star} \tilde{v}_{e,a_e}^{-1} \leq \Delta(a)^2, \quad \forall a \in \mathcal{A}. \quad (3)$$

The proof is presented in App. A.2. This approximation reduces the number of variables from $K^N$ to $\tilde{A} + KN$. The lemma states that, for acyclic factor graphs, the locally tree-like approximation (3) is tight, i.e., $C_\theta^L = C_\theta^\star$. Unfortunately, for general graphs, we have that $C_\theta^L < C_\theta^\star$ (a direct consequence of [39, Prop. 4.1]), and hence it is impossible to devise algorithms based on this approximation.

### 5.1.2 $\psi$-Mean-Field reduction

Our $\psi$-MF reduction is loosely inspired by MF approximation methods in graphical models [39]. It consists in decomposing global variables $v$ as a function $\psi$ of the local variables $w = (w_i)_{i \in [N]}$. Specifically, the $\psi$-MF reduction introduces the following set of constraints: $v_a = \psi_a(w), \forall a \neq a_\theta^\star$, where $\psi_a : \mathbb{R}_{\geq 0}^{KN} \to \mathbb{R}_{\geq 0}$. Let $\mathcal{V}_\psi = \left\{ v \in \mathbb{R}_{\geq 0}^A : \exists w \in \mathbb{R}_{\geq 0}^{KN}, v_a = \psi_a(w), \forall a \neq a_\theta^\star \right\}$, and define the $\psi$-MF marginal cone as

$$\tilde{\mathcal{V}}_{\psi\text{-MF}} = \left\{ \tilde{v} \in \mathbb{R}_{\geq 0}^{\tilde{A}} : \exists v \in \mathcal{V}_\psi, \forall e \in [\rho], a_e \in \mathcal{A}_e, \tilde{v}_{e,a_e} = \sum_{b \in \mathcal{A} \setminus \{a_\theta^\star\} : b_e = a_e} v_b, \right\}.$$

We get the $\psi$-MF approximation, $C_\theta^{\psi\text{-MF}}$ by replacing $\tilde{\mathcal{V}}$ by $\tilde{\mathcal{V}}_{\psi\text{-MF}}$ in (2).

**Lemma 5.2.** *For any $\theta, \psi$, we have that $C_\theta^\star \leq C_\theta^{\psi\text{-MF}}$, where $C_\theta^{\psi\text{-MF}}$ is the value of the optimization problem:*

$$\min_{\tilde{v} \in \tilde{\mathcal{V}}_{\psi\text{-MF}}} \sum_{e \in [\rho], a_e \in \mathcal{A}_e} \tilde{v}_{e,a_e}(\theta_e(a_e^\star) - \theta_e(a_e)) \quad \text{s.t.} \quad \sum_{e \in [\rho]: a_e \neq a_e^\star} \tilde{v}_{e,a_e}^{-1} \leq \Delta(a)^2, \quad \forall a \in \mathcal{A}. \quad (4)$$

Clearly, the tractability of the problem $C_\theta^{\psi\text{-MF}}$ depends on the choice of $\psi$. A natural choice would be $\psi_a(w) = \prod_{i \in [N]} w_{i,a_i}$, as proposed, e.g., in approximate inference methods [39]. However, this choice leads to a non-convex program (see App. B). The following lemma proposes a choice of $\psi$ which leads to a convex program over local variables $w$ only.

**Lemma 5.3.** *Let $\psi_a(w) = \sum_{i \in [N]} w_{i,a_i}, \forall a \neq a^\star$. Then $C_\theta^{\psi\text{-MF}}$ is the value of the following convex optimization problem:*

$$\min_{w \in \mathbb{R}_{\geq 0}^{KN}} \sum_{e \in [\rho], a_e \in \mathcal{A}_e} f_{e,a_e}(w)(\theta(a_e^\star) - \theta(a_e)) \quad \text{s.t.} \quad \sum_{e \in [\rho]: a_e \neq a_e^\star} f_{e,a_e}(w)^{-1} \leq \Delta(a)^2, \forall a \in \mathcal{A}, \quad (5)$$

*where $f_{e,a_e}(w) = K^{N-|\mathcal{S}_e|} \sum_{i \in \mathcal{S}_e, a_i \in \mathcal{A}_i} w_{i,a_i} + K^{N-|\mathcal{S}_e|-1} \sum_{i \notin \mathcal{S}_e} w_{i,a_i}$. Furthermore, it holds that $C_\theta^{\psi\text{-MF}} \leq \rho \Delta_{\min}^{-2} \sum_{e \in [\rho], a_e \in \mathcal{A}_e} (\theta(a_e^\star) - \theta_e(a_e))$, where $\Delta_{\min} = \min_{a \neq a^\star} \Delta(a)$.*

The proof is presented in App. A.3. The lemma provides a worst-case scaling of $C_\theta^{\psi\text{-MF}}$: it scales at most as $\tilde{A} = \sum_{e \in [\rho]} K^{|\mathcal{S}_e|}$ (remember that if we were considering a MAMAB as a standard bandit problem, the latter would have $K^N$ arm and hence a regret scaling exponentially in $N$). The number of variables involved in (5) is $KN$. The quantities $(f_{e,a_e}(w))_{e \in [\rho], a_e \in \mathcal{A}_e}$ are group quantities interpreted as the group variables $\tilde{v}_{e,a_e}$, and uniquely determined by local variables $w$. In the following, we use the notation $C_\theta^{\text{MF}}$ to represent $C_\theta^{\psi\text{-MF}}$ for the function $\psi$ defined in Lemma 5.3.

## 5.2 Constraint reduction

The remaining challenge is to reduce the number of constraints in (3), (4) or (5). For each global action $a$, the constraint writes $\sum_{e \in [\rho]} \tilde{v}_{e,a_e}^{-1} \leq \Delta(a)^2$. The major issue is the non-linearity of the function appearing in the constraints w.r.t. group actions $a_e$. Upon inspection, it appears that the heterogeneity in the gaps (generally $\Delta(a) \neq \Delta(b)$ for $a \neq b$) is causing the non-linearity. To address this problem, we present, in the following lemma, a family of approximations leveraging an ordering of the first $m$ smallest gaps. For $m \in [K^N]$, let $a^{(m)}$ be the $m^{\text{th}}$ best global action and, for $m \in [K^N - 1]$, let $\Delta_m = \theta(a_\theta^\star) - \theta(a^{(m+1)})$ be the $m^{\text{th}}$ minimal non-zero gap (with ties breaking arbitrarily).

**Lemma 5.4.** *Let $m \in [K^N - 1]$, and $\diamond \in \{\mathrm{L}, \mathrm{MF}\}$. Let $C_\theta^\diamond(m)$ be the value of the convex program:*

$$\min_{\tilde{v} \in \tilde{\mathcal{V}}_\diamond} \sum_{e \in [\rho], a_e \in \mathcal{A}_e} \tilde{v}_{e,a_e} (\theta_e(a_e^\star) - \theta_e(a_e)) \tag{6}$$

$$\text{s.t.} \sum_{e \in [\rho]: a_e^{(j+1)} \neq a_e^\star} \tilde{v}_{e,a_e^{(j+1)}}^{-1} \leq \Delta_j^2, \quad \forall j \in [m] \tag{7}$$

$$\sum_{e \in [\rho]: a_e \neq a_e^\star} \tilde{v}_{e,a_e}^{-1} \leq \Delta_m^2, \quad \forall a \in \mathcal{A} \setminus \cup_{j \in [m]} \{a^{(j+1)}\}. \tag{8}$$

*Then, for any $\diamond \in \{\mathrm{L}, \mathrm{MF}\}$, $m \in [K^N - 2]$, we have $C_\theta^\diamond(m+1) \leq C_\theta^\diamond(m)$, $C_\theta^\star \leq C_\theta^\diamond(m)$, and by definition $C_\theta^\diamond(K^N - 1) = C_\theta^\diamond$.*

The proof is reported in App. F. Clearly, $C_\theta^\diamond(m)$ has still $|\mathcal{A}|$ constraints (7)-(8). However, as the gap $\Delta_m$ used in (8) is constant, these constraints are now a linear sum of terms depending on group actions $a_e$. For constraints with this type of structure, there exists an efficient and provably equivalent representation. The procedure yielding this representation, which we refer to as FCR, is based on a generalization of the popular Factored LP algorithm described in [19, 20] for Factored Markov Decision Processes (FMDPs).

For the sake of brevity, we briefly describe the procedure below and postpone its detailed exposition to App. G. FCR is inspired by the Variable Elimination (VE) procedure in graphical models [14]. It iteratively eliminates constraints from (8), according to an elimination order $\mathcal{O}$. The elimination procedure induces an *elimination graph*, which encodes dependencies between constraints as we perform elimination. As shown in the following lemma, the number of constraints is exponential in the degree $A_\mathcal{O}$ of the elimination graph induced by the order of elimination $\mathcal{O}$.

**Lemma 5.5.** *There exists a procedure which, given the constraints in (8) returns a provably equivalent constraint set of size $O(NK^{A_\mathcal{O}+1})$.*

Although for general graphs finding an ordering $\mathcal{O}$ minimizing $A_\mathcal{O}$ is an $\mathcal{NP}$-hard problem [14], for specific graphs there are orderings yielding $A_\mathcal{O} \ll N$. For example, these orderings yield $A_\mathcal{O} = 2$ for line or star factor graphs, and $A_\mathcal{O} = 3$ for ring factor graphs, independently of the number of agents $N$ (see Fig. 3 and refer to App. J for details). Solving $C_\theta^\diamond(m)$, requires computing the first $m + 1$ best global actions and the $m$ minimal gaps. To solve this task, the *elim-m-opt* algorithm [15] has complexity $O((m + 1)NK^{A_\mathcal{O}+1})$ (see App. G). Fig. 2 shows an illustration of the trade-off between statistical and computational complexity. Note that ESM is meant to be applied when $m$ does not grow exponentially in $N$. In practice, we observed that selecting $m = \tilde{A}$ yields a good trade-off between statistical complexity and computational complexity.

## 6 The ESM algorithm

In this section, we present ESM, an algorithm whose regret matches (asymptotically) our approximated lower bounds. The algorithm is inspired by OSSB [8]. It ensures that sub-optimal actions are sampled as prescribed by the solution of the approximated optimization problems $C_\theta^{\mathrm{MF}}(m)$ or $C_\theta^{\mathrm{L}}(m)$: each group $e$ must explore each group action $a_e$ for $\tilde{v}_{e,a_e} \log(T)$ times, and selecting the action yielding the largest estimated reward for the remaining rounds. As the lower bounds depend on the unknown parameter $\theta$, it has to be estimated.

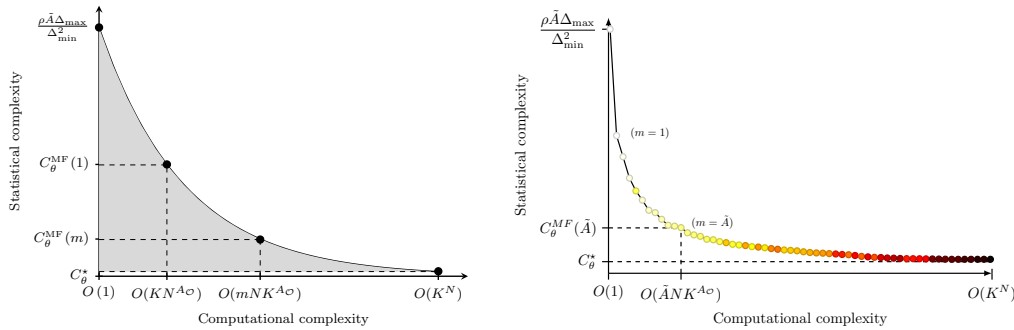

Figure 2: Left: idealized curve illustrating possible ranges of the trade-off between statistical and computational complexity for $C^{\mathrm{MF}}(m)$, when varying $m$. Right: an instance of this trade-off for a line factor graph (darker colors for the points represent higher running times). Selecting $m = \tilde{A}$ yields a good trade-off between computational and statistical complexity for this instance.

Generally, the estimation of $\theta$ would require evaluating an exponentially large number of components, i.e., $\theta(a), \forall a \in \mathcal{A}$. Instead, by leveraging the factored reward structure, we can simply focus on estimating group parameters $(\theta_e)_{e\in[\rho]}$. We define the estimate at time $t$, group $e$, and action $a_e$ as $\hat{\theta}_{t,e,a_e} = \frac{1}{N_{t,e,a_e}} \sum_{s\in[t]:a_{s,e}=a_e} r_{s,e}$ where $N_{t,e,a_e} = \sum_{s\in[t]} \mathbb{1}_{\{a_{s,e}=a_e\}}$ is the number of times action $a_e$ is selected for group $e$. We also define $\hat{\theta}_t = (\hat{\theta}_{t,e,a_e})_{e\in[\rho],a_e\in\mathcal{A}_e}$, and $N_t = (N_{t,e,a_e})_{e\in[\rho],a_e\in\mathcal{A}_e}$.

---

**Algorithm 1** ESM($\mathcal{A}_0, \varepsilon, \gamma, \diamond, m$)

---

   Sample each group actions in $\mathcal{A}_0$ once and update $(N_{T_0}, \hat{\theta}_{T_0})$; $s_{T_0} = 0$       ▷ **Initialization**
   **for** $t = T_0, \ldots, T$ **do**
      $\left((\tilde{v}_{t,e})_{e\in[\rho]}\right) \leftarrow$ Solve $C^{\diamond}_{\hat{\theta}_t}(m)$
      **if** $N_{t,e,a_e} \geq (1+\gamma)\tilde{v}_{t,e,a_e} \log(t), \forall e \in [\rho], a_e \in \mathcal{A}_e$ **then**      ▷ **Exploitation**
         $a_t = a^{\star}_{\hat{\theta}_t}$
         $s_t = s_{t-1}$
      **else**
         $s_t = s_{t-1} + 1$
         **if** $\min_{e\in[\rho],a_e\in\mathcal{A}_e} N_{t,e,a_e} \leq \varepsilon s_t$ **then**      ▷ **Estimation**
            $a_t \in \mathcal{A}_0 : a_{t,e'} = b_{e'}$ with $(e', b_{e'}) \in \arg\min_{e,a_e} N_{t,e,a_e}$
         **else**      ▷ **Exploration**
            $a_t \in \mathcal{A} : a_{t,e'} = b_{e'}$ with $(e', b_{e'}) \in \arg\min_{e,a_e} \frac{N_{t,e,a_e}}{\tilde{v}_{t,e,a_e}}$
   Update $(N_{t,e,a_{t,e}}, \hat{\theta}_{t,e,a_{t,e}})_{e\in[\rho]}$

---

The pseudocode of ESM is presented in Alg. 1. It takes as inputs two exploration parameters $\varepsilon, \gamma > 0$, an exploration set $\mathcal{A}_0 \subset \mathcal{A}$, the approximation parameter $m \in [K^N - 1]$, and $\diamond \in \{\mathrm{MF}, \mathrm{L}\}$ depending on the targeted regret lower bound approximation. The parameters $\varepsilon, \gamma > 0$ impact the amount of exploration performed by ESM. When decreasing both these parameters the exploration of ESM also decreases. After an initialization phase, the algorithm alternates between three additional phases as described below.

**Initialization.** In the initialization phase, we select actions from $\mathcal{A}_0$ to ensure that each group action is sampled at least once. The set $\mathcal{A}_0 \subseteq \mathcal{A}$ is chosen in such a way that it covers all possible group actions, i.e., $\mathcal{A}_0$ is such that $\forall e \in [\rho], \forall a_e \in \mathcal{A}_e, \exists b \in \mathcal{A}_0 : b_e = a_e$. In App. I, we present an efficient routine to select $\mathcal{A}_0$. Let $T_0 = \inf\{t \geq 0 : N_{t,e,a_e} > 0, \forall e \in [\rho], a_e \in \mathcal{A}_e\}$ be the length of the initialization phase. For $t \leq T_0$ we select $a_t \in \mathcal{A}_0 : a_{t,e'} = b_{e'}$ with $(e', b_{e'}) \in \arg\min_{e,a_e} N_{t,e,a_e}$ (with ties breaking arbitrarily), i.e., we select a global action containing the most under-explored group action. This choice ensures that $T_0 \leq \tilde{A}$. For $t > T_0$, the algorithm solves the approximated lower bound optimization problem $C^{\diamond}_{\hat{\theta}_t}(m)$ and alternates between exploitation, exploration, and estimation.

**Exploitation.** If $N_{t,e,a_e} \geq (1 + \gamma)\tilde{v}_{t,e,a_e} \log(t)$, ESM enters the exploitation phase: it selects the best empirical action $a^\star_{\hat{\theta}_t} = \arg\max_{a \in \mathcal{A}} \sum_{e \in [\rho]} \hat{\theta}_{t,e,a_e}$. Generally, computing $a^\star_{\hat{\theta}_t}$ requires a $\max$ operation over an exponential number of actions $a \in \mathcal{A}$. Fortunately, due to the factored structure, we can implement the $\max$ operation efficiently through a VE procedure [39] (see App. G).

**Estimation.** If not enough information has been gathered, ESM enters an estimation phase, where it selects the least explored group action similarly to the initialization phase. This ensures that the *certainty equivalence* holds, i.e., that $\hat{\theta}_t$ is estimated accurately.

**Exploration.** Otherwise, the algorithm enters the exploration phase and selects actions as suggested by the solution of $C^\diamond_{\hat{\theta}_t}(m)$. More precisely, we select a global action $a_t \in \mathcal{A}$ which contains a group action $a_{t,e'} = b_{e'}$ that minimizes the following ratio where $e'$ and $b_{e'}$ are the group index and group action which minimize $\frac{N_{t,e,a_e}}{\tilde{v}_{t,e,a_e}}$.

**Upper bound.** We establish that the ESM algorithm achieves a regret, matching the approximate lower bound $C^\diamond_\theta(m) \log(T)$, asymptotically as $T \to \infty$. The proof is given in App. D.

**Theorem 6.1.** *Let $\varepsilon < 1/|\mathcal{A}_0|$. For any $m \in [K^N - 1]$, we have that*

1. *$\limsup_{T \to \infty} \frac{R^\pi(T)}{\log(T)} \leq C^{\mathrm{MF}}_\theta(m)\xi(\varepsilon, \gamma)$, for $\pi = \mathrm{ESM}(\mathcal{A}_0, \varepsilon, \gamma, \mathrm{MF}, m)$, for any $\theta$,*

2. *$\limsup_{T \to \infty} \frac{R^\pi(T)}{\log(T)} \leq C^{\mathrm{L}}_\theta(m)\xi(\varepsilon, \gamma)$, for $\pi = \mathrm{ESM}(\mathcal{A}_0, \varepsilon, \gamma, \mathrm{L}, m)$, for any $\theta$ described by acyclic factor graphs,*

*where $\xi$ is a function such that $\lim_{(\varepsilon,\gamma) \to (0,0)} \xi(\varepsilon, \gamma) = 1$.*

## 7  Experiments

In this section, we present numerical experiments to assess the performance of our algorithm. We propose two sets of experiments: $(i)$ a set of synthetic MAMABs with different graph topologies, and $(ii)$ an industrial use-case from the radio communication domain: *antenna tilt optimization*. The code for the synthetic experiments and the additional experiments presented in App. K is available at this link.

### 7.1  Synthetic Experiments

**Problem instances.** We consider the factor graphs depicted in Fig. 3. The expected rewards are selected uniformly at random in the interval $[0, 10]$. In our experiments, select $N = 5$ and $K = 3$. We execute our experiments for $N_{\mathrm{sim}} = 5$ independent runs. Following previous work [8], we select $\gamma = 0$, and $\varepsilon = 0.01$. The elimination order is chosen as $\mathcal{O} = [N]$. We implement the solver for the lower bound optimization problems using CVXPY [17] with a MOSEK solver [1].

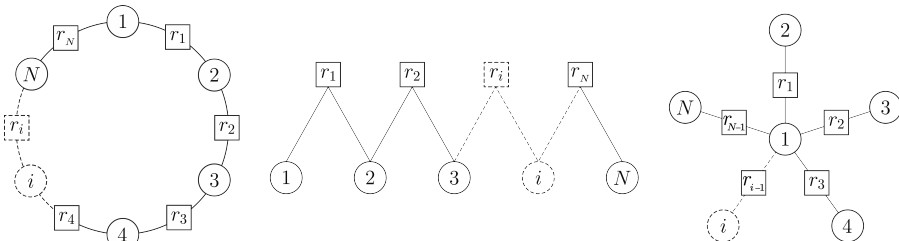

Figure 3: Factor graphs used in the synthetic experiments: *ring* (left), *line* (center), *star* (right).

**Results.** The results for the regret (in log scale) are presented in Fig. 4. The performance of ESM is compared to that of MAUCE [3], HEIST [35], and to a random strategy selecting actions uniformly at random. The computational complexity results, reported in Fig. 5, measure the running time (in sec.) to solve an instance of the approximate lower bound optimization problem $C^\diamond_\theta(m)$. We use $m = \tilde{A}$, and $\diamond = \mathrm{MF}$ for the ring, or $\diamond = \mathrm{L}$ for the star and line graph topologies.

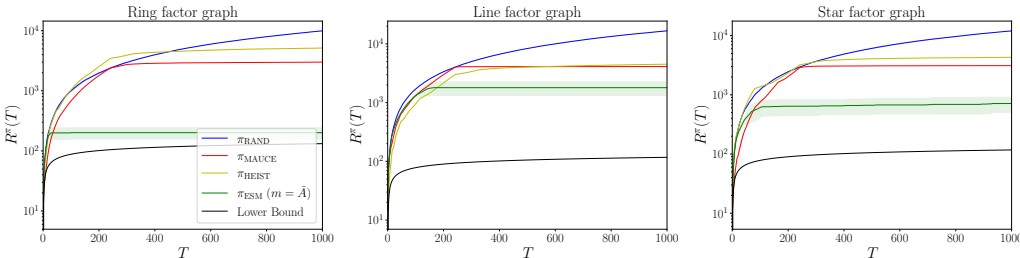

Figure 4: Regret results for the synthetic instances.

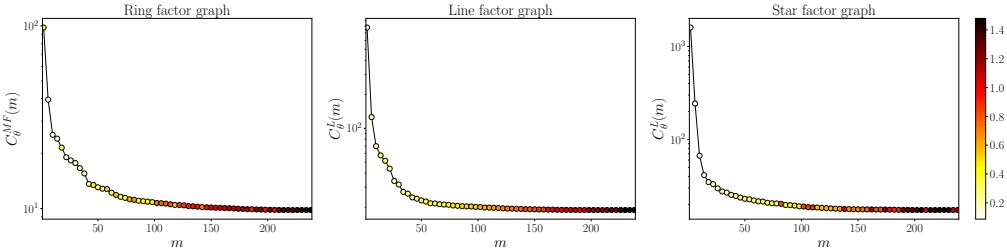

Figure 5: Running time (in sec.) to solve an instance of $C_\diamond^\theta(m)$, when varying $m$.

## 7.2 Antenna tilt optimization

Next, we test our algorithm on a radio network optimization task. The goal is to control the vertical antenna tilt at different network Base Stations to optimize the network throughput. In the following, we detail the network model, our simulation setup, and present our experimental results.

**Network model.** We consider a sectorized radio network consisting of a set of *sectors* $\mathcal{S} = [N]$. The set of sectors corresponds to the set of agents in our MAMAB framework. Since each sector is associated to a unique antenna, we will use the terms *sector* and *antenna* interchangeably. We assume that each sector $i \in \mathcal{S}$ serves (on the downlink) a fixed set of Users Equipments (UEs) $\mathcal{U}_i$ (each UE is associated with a unique antenna, that from which it receives the strongest signal).

**Factor graph.** We model the observed reward in the radio network as a factor graph with $N = |\mathcal{S}|$ agent nodes and $\rho = |\mathcal{S}|$ factor nodes. Each sector is associated with a unique factor, which models the rewards observed in that sector. We build the factor graph based on the interference pattern of the antennas, i.e., antennas that can interfere with each other are connected to common factors. An example of such a graph and additional experimental details are reported in App. L.

**Actions and rewards.** The action $a_{t,i}$ represents the antenna tilt for sector $i \in \mathcal{S}$ and at time $t$. It is chosen from a discrete set of $K$ tilts, i.e., $a_{t,i} \in \{\alpha_1, \ldots, \alpha_K\}$. The tilt for a group of sectors $e$ is denoted by $a_e$. Rewards are based on the throughput of UEs in sector $i$, which depends on the actions of a group of agents $a_e$: $r_e(a_e) = \sum_{u \in \mathcal{U}_i} T_{i,u}(a_e)$, where $T_{i,u}$ is the throughput of an UE $u$ associated to sector $i$. Hence, the global reward for a tilt configuration $a \in \mathcal{A}$ is $r(a) = \sum_{i \in [N]} \sum_{u \in \mathcal{U}_i} T_{i,u}(a_e)$. The throughput $T_{i,u}$ depends on channel conditions (or *fading*) between the antenna and the user. These conditions rapidly evolve over time around their mean.

**Simulator.** We run our experiments in a proprietary mobile network simulator in an urban environment. The simulation parameters used in our experiments are reported in App. K. Based on the user positions and network parameters, the simulator computes the path loss in the network environment using a BEZT propagation model [32] and returns the throughput for each sector by conducting user association and resource allocation in a full-buffer traffic demand scenario.

**Results.** We test our algorithm for $\mathcal{A}_i = \{2°, 7°, 13°\}$, and for $|\mathcal{S}| = 6$ sectors. As the factor graph contains cycles, we use $\diamond = \text{MF}$ and select $m = 3$. The results, presented in Fig. 6, are in line with the experimental findings of the previous section. However, the ESM running time is higher due to the higher complexity of the factor graph.

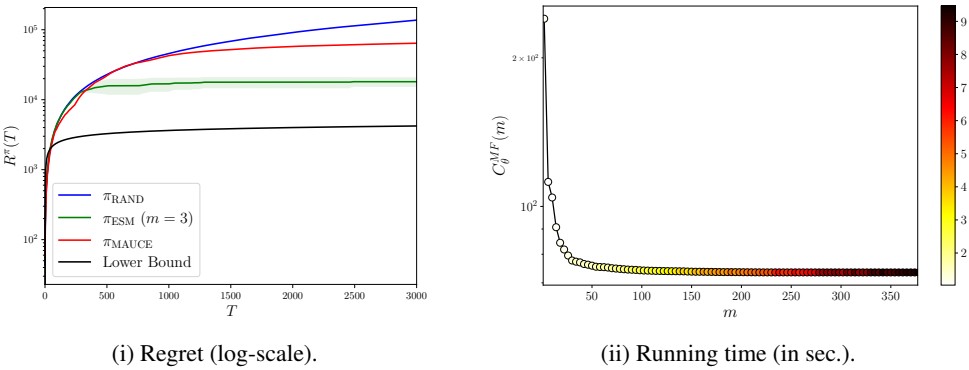

(i) Regret (log-scale).

(ii) Running time (in sec.).

Figure 6: Results for the antenna tilt optimization experiments.

# 8    Conclusions

In this paper, we investigated the problem of regret minimization in MAMABs: we derived a regret lower bound, proposed approximations of it, and devised ESM, an algorithm trading off statistical and computational efficiency. We then assessed the performance of ESM on both synthetic examples and the antenna tilt optimization problem. Interesting future research directions include proposing efficient distributed implementations of ESM, quantifying on its communication complexity, and investigating representation learning problems in MAMABs where the underlying factor graph defining the reward is unknown and needs to be learned.

## Acknowledgement

This research was supported by the Wallenberg AI, Autonomous Systems and Software Program (WASP) funded by the Knut and Alice Wallenberg Foundation.

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
