# Appendix

# Contents

# A  Lower bound proofs

In this section, we present proofs for the regret lower bound (Th. 4.1) and for the lower bound variable reduction results (Lemma 5.1, Lemma 5.2, Lemma 5.3). The lower bound proof, presented in App. A.5, is a direct consequence of an analogous bound in the combinatorial semi-bandit feedback setting, first given in [10, Th. 1] and leverages general techniques for controlled Markov chains [18]. The approximations leverage methods for approximate inference in probabilistic graphical models [39, 27]. In addition, we presenting an alternative characterization of the lower bound in App. A.5.

## A.1  Proof of Theorem 4.1

*Proof.* Let $\Theta = \{\theta \in \mathbb{R}^A : \exists (\theta_e(a_e))_{e \in [\rho], a_e \in \mathcal{A}}, \forall a \in \mathcal{A}, \theta(a) = \sum_{e \in [\rho]} \theta_e(a_e), a_\theta^\star \text{ is unique}\}$. As shown in App. E, the MAMAB problem is a particular case of combinatorial linear bandits with semi-bandit feedback. By interpreting the MAMAB problem in this setting, and using the Gaussian reward assumption, the result from [10, Th. 1] implies that $C_\theta^\star$ is the value of the following semi-infinite optimization problem:

$$\min_{v \in \mathbb{R}_{\geq 0}^A, \tilde{v} \in \tilde{\mathcal{V}}} \quad \sum_{a \in \mathcal{A}} v_a \Delta(a) \tag{9}$$

$$\text{subject to} \quad \inf_{\lambda \in B(\theta)} \left\{ \sum_{e \in [\rho]} \sum_{a_e \in \mathcal{A}_e} \tilde{v}_{e,a_e} (\theta_e(a_e) - \lambda_e(a_e))^2 \right\} \geq 1, \tag{10}$$

where $\tilde{v}_{e,a_e} = \sum_{b \in \mathcal{A}: a_e = b_e} v_b$, and the set of confusing parameters is defined as (here we use $a^\star = a_\theta^\star$ for conciseness)

$$B(\theta) = \left\{ \lambda \in \Theta : \lambda_e(a_e^\star) = \theta_e(a_e^\star), \forall e \in [\rho] \text{ and } \exists a \neq a^\star : \sum_{e \in [\rho]} \lambda_e(a_e) - \lambda_e(a_e^\star) > 0 \right\},$$

Let us introduce for $\kappa > 0$, the set

$$B_\kappa(\theta) = \left\{ \lambda \in \Theta : \lambda_e(a_e^\star) = \theta_e(a_e^\star), \forall e \in [\rho] \text{ and } \exists a \neq a^\star : \sum_{e \in [\rho]} \lambda_e(a_e) - \lambda_e(a_e^\star) \geq \kappa \right\}.$$

The set $B_\kappa$ can be decomposed as

$$B(\theta) = \bigcup_{\kappa > 0} \bigcup_{a \neq a_\theta^\star} \underbrace{\left\{ \lambda \in B(\theta) : \sum_{e \in [\rho]} \lambda_e(a_e) - \theta_e(a_e^\star) \geq \kappa \right\}}_{\triangleq B_{\kappa,a}(\theta)}, \tag{11}$$

and we have that The LHS of the optimization problem defining the constraints in (10) can be rewritten as

$$\inf_{\lambda \in B_\kappa(\theta)} \sum_{e \in [\rho], a_e \in \mathcal{A}_e} \tilde{v}_{e,a_e} (\theta_e(a_e) - \lambda_e(a_e))^2 = \min_{\kappa > 0} \min_{a \neq a^\star} \inf_{\lambda \in B_{\kappa,a}(\theta)} \sum_{e \in [\rho], a_e \in \mathcal{A}_e} \tilde{v}_{e,a_e} (\theta_e(a_e) - \lambda_e(a_e))^2.$$

Now, the optimization problem $\inf_{\lambda \in B_{\kappa,a}(\theta)} \sum_{e \in [\rho], a_e \in \mathcal{A}_e} \tilde{v}_{e,a_e} (\theta_e(a_e) - \lambda_e(a_e))^2$ is convex and satisfies Slater's conditions [4]. By solving the KKT system, and letting $\kappa \to 0$, it is possible to verify that:

$$\inf_{\lambda \in B_{\kappa,a}(\theta)} \sum_{e \in [\rho], a_e \in \mathcal{A}_e} \tilde{v}_{e,a_e} (\theta_e(a_e) - \lambda_e(a_e))^2 \xrightarrow[\kappa \to 0]{} \frac{\sum_{e \in [\rho]: a_e \neq a_e^\star} \tilde{v}_{e,a_e}^{-1}}{\Delta(a)^2}.$$

The result follows directly by substituting the solution into the constraint on the LHS of 10, and by noting that the objective (9) can be rewritten, $\forall \tilde{v} \in \tilde{\mathcal{V}}$, as

$$\sum_{a \in \mathcal{A}} v_a \Delta(a) = \sum_{e \in [\rho], a_e \in \mathcal{A}_e} \tilde{v}_{e,a_e} (\theta_e(a_e^\star) - \theta_e(a_e)).$$

$\square$

## A.2 Proof of Lemma 5.1

*Proof.* The proof is a consequence of a fundamental result in probabilistic graphical models. More precisely, it stems from the equivalence between the so-called *marginal polytope* and *local polytope* described in [39, Prop. 4.1], for tree-structured factor graphs, described in the following paragraph.

Let $\Lambda = \left\{ \lambda \in [0,1]^{|\mathcal{A}|} : \sum_{a \in \mathcal{A}} \lambda_a = 1 \right\}$ be the $|\mathcal{A}| - 1$-dimensional simplex. Define the *marginal polytope* as

$$\tilde{\Lambda} = \left\{ \tilde{\lambda} \in \mathbb{R}_{\geq 0}^{\tilde{A}} : \exists \lambda \in \Lambda : \forall e \in [\rho], a_e \in \mathcal{A}_e, \tilde{\lambda}_{e,a_e} = \sum_{b \in \mathcal{A}: b_e = a_e} \lambda_b \right\}, \tag{12}$$

and the *local polytope* is

$$\tilde{\Lambda}_L = \left\{ \tilde{\lambda} \in \mathbb{R}_{\geq 0}^{\tilde{A}} : \exists l \in [0,1]^{KN}, \forall e \in [\rho], i \in \mathcal{S}_e, \forall a_i \in \mathcal{A}_i, l_{i,a_i} = \sum_{b_e \in \mathcal{A}_e : b_e \sim a_i} \tilde{\lambda}_{e,b_e} \right\}.$$

The following lemma states that the marginal polytope is an outer approximation of the local polytope, and that for acyclic factor graphs (see App. J for a formal definition), these two sets are equivalent.

**Lemma A.1** (Prop. 4.1, [39]). *The inclusion $\tilde{\Lambda} \subseteq \tilde{\Lambda}_L$ holds for any factor graph. Additionally, for acyclic factor graphs, we have that $\tilde{\Lambda} = \tilde{\Lambda}_L$.*

The sets $\tilde{\Lambda}$ and $\tilde{\Lambda}_L$ are essentially equivalent to the *marginal cone* $\tilde{\mathcal{V}}$ and *local cone* $\tilde{\mathcal{V}}_L$, respectively, when including the additional unitary constraint which enforces that the sum of the variables involved in the sets is 1. Lemma A.1, automatically implies that $\mathcal{V} = \mathcal{V}_L$, for MAMABs whose underlying factor graph is acyclic, and hence $C_\theta^\star = C_\theta^L$ holds. Instead, for MAMABs with generic factor graphs Lem A.1 implies that $\tilde{\mathcal{V}} \subseteq \tilde{\mathcal{V}}_L$ and hence:

$$\left\{ \tilde{v} \in \tilde{\mathcal{V}} : \frac{\sum_{e \in [\rho]: a_e \neq a_e^\star} \tilde{v}_{e,a_e}^{-1}}{\Delta(a)^2} \leq 1, \forall a \neq a^\star \right\} \subseteq \left\{ \tilde{v} \in \tilde{\mathcal{V}}_L : \frac{\sum_{e \in [\rho]: a_e \neq a_e^\star} \tilde{v}_{e,a_e}^{-1}}{\Delta(a)^2} \leq 1, \forall a \neq a^\star \right\}.$$

Hence, the bound $C_\theta^L \leq C_\theta^\star$ follows by noting that for a function $f$ and a pair of sets $(\mathcal{X}, \mathcal{X}')$ such that $\mathcal{X} \subseteq \mathcal{X}'$, we have that $\min_{x \in \mathcal{X}'} f(x) \leq \min_{x \in \mathcal{X}} f(x)$ (see e.g. [4]). $\qquad \square$

## A.3 Proof of Lemma 5.2

*Proof.* Let $\psi$ be any function. Recall the definitions of

$$\mathcal{V}_\psi = \left\{ v \in \mathbb{R}_{\geq 0}^A : \exists w \in \mathbb{R}_{\geq 0}^{KN}, v_a = \psi_a(w), \forall a \neq a_\theta^\star \right\}.$$

The $\psi$-MF cone is then defined as:

$$\tilde{\mathcal{V}}_{\psi\text{-MF}} = \left\{ \tilde{v} \in \mathbb{R}_{\geq 0}^{\tilde{A}} : \exists v \in \mathcal{V}_\psi, \forall e \in [\rho], a_e \in \mathcal{A}_e, \tilde{v}_{e,a_e} = \sum_{b \in \mathcal{A} \setminus \{a_\theta^\star\}: b_e = a_e} v_b, \right\}.$$

It is immediate to check that for any $\psi$, we have that $\tilde{\mathcal{V}}_{\psi\text{-MF}} \subseteq \tilde{\mathcal{V}}$, and hence

$$\left\{ \tilde{v} \in \tilde{\mathcal{V}}_{\psi\text{-MF}} : \frac{\sum_{e \in [\rho]: a_e \neq a_e^\star} \tilde{v}_{e,a_e}^{-1}}{\Delta(a)^2} \leq 1, \forall a \neq a^\star \right\} \subseteq \left\{ \tilde{v} \in \tilde{\mathcal{V}} : \frac{\sum_{e \in [\rho]: a_e \neq a_e^\star} \tilde{v}_{e,a_e}^{-1}}{\Delta(a)^2} \leq 1, \forall a \neq a^\star \right\}.$$

By a similar argument to Lemma 5.1 we conclude that $C_\theta^\star \leq C_\theta^{\psi\text{-MF}}$. $\qquad \square$

## A.4 Proof of Lemma 5.3

*Proof.* We first show that $C_\theta^{\text{MF}} = C_\theta^{\psi\text{-MF}}$ can be written in terms of $w$ only for the choice of $\psi$ such that $\psi_a(w) = \sum_{i\in[N]} w_{i,a_i}, \forall a \in \mathcal{A}$. Let us denote $\tilde{\mathcal{V}}_{\psi\text{-MF}} = \tilde{\mathcal{V}}_{\text{MF}}$. By definition, we can check that any $\tilde{v} \in \tilde{\mathcal{V}}_{\text{MF}}$ satisfies

$$\tilde{v}_{e,a_e} = \sum_{b\in\mathcal{A}:b_e=a_e} w_b = \sum_{b\in\mathcal{A}:b_e=a_e} \sum_{i\in[N]} w_{i,b_i} = K^{N-|\mathcal{S}_e|} \sum_{i\in\mathcal{S}_e} w_{i,a_i} + K^{N-|\mathcal{S}_e|-1} \sum_{i\notin\mathcal{S}_e, a_i\in\mathcal{A}_i} w_{i,a_i}. \tag{13}$$

As $\tilde{v}$ can be uniquely written in terms of local variables $(w_i)_{i\in[N]}$, by substituting (13) it into (5), note that the resulting optimization problem involves only local variables.

Next, we show that, for any $\theta$ and $\psi$, $C_\theta^{\psi\text{-MF}} \le \rho\Delta_{\min}^{-2} \sum_{e\in[\rho],a_e\in\mathcal{A}_e}(\theta_e(a_e^\star) - \theta_e(a_e))$. The bound is obtained by adapting [12, Lemma 9], [10, Corollary 1]. First, note that $\tilde{v} \in \tilde{\mathcal{V}}$ such that $\tilde{v}_{e,a_e} = \frac{\rho}{\Delta_{\min}^2}$, $\forall e \in [\rho], a_e \in \mathcal{A}_e$ is a feasible solution to (1), as for this $\tilde{v}$, it holds that

$$\sum_{e\in[\rho]:a_e\neq a_e^\star} \tilde{v}_{e,a_e}^{-1} = \frac{\Delta_{\min}^2}{\rho} \sum_{e\in[\rho]} \mathbb{1}_{\{a_e\neq a_e^\star\}} \le \Delta_{\min}^2 \le \Delta(a)^2.$$

Hence, by substitution into the objective of (5), we get the result:

$$\sum_{e\in[\rho],a_e\in\mathcal{A}_e} \tilde{v}_{a_e}(\theta_e(a_e^\star) - \theta_e(a_e)) \le \frac{\rho\sum_{e\in[\rho],a_e\in\mathcal{A}_e}(\theta_e(a_e^\star) - \theta_e(a_e))}{\Delta_{\min}^2}.$$

$\square$

## A.5 Reformulation of the lower bound

The following lemma provides an alternative characterization of the lower bound (1) in terms of a min-max optimization problem with variables over the marginal polytope $\tilde{\Lambda}$ introduced in App. A.2. The reformulation is similar to the analogous one in the plain-bandit literature [23].

**Lemma A.2.** *For any $\theta$, $C_\theta^\star$ is the value of the following convex optimization problem*

$$C_\theta^\star = \inf_{\tilde{\lambda}\in\tilde{\Lambda}} \max_{a\neq a_\theta^\star} \frac{\sum_{e\in[\rho]:a_e\neq a_e^\star} \tilde{\lambda}_{e,a_e}^{-1}}{\Delta(a)^2} \sum_{e\in[\rho],a_e\in\mathcal{A}_e} \tilde{\lambda}_{e,a_e}(\theta_e(a_e^\star) - \theta_e(a_e)). \tag{14}$$

*Proof.* The proof follows similar steps to [23, Thm. 1.8], which proves the result in the plain bandit setting. Define the set of feasible solutions for the lower bound problem (1) as:

$$\mathcal{F}(\theta) = \left\{ \tilde{v} \in \tilde{\mathcal{V}} : \frac{\sum_{e\in[\rho]:a_e\neq a_e^\star} \tilde{v}_{e,a_e}^{-1}}{\Delta(a)^2} \le 1, \forall a \neq a^\star \right\},$$

Note that, if $\tilde{\lambda} \in \tilde{\Lambda}$, any $\tilde{v}$ such that $\forall a \neq a_\theta^\star$, $\tilde{v}_{e,a_e} = \frac{\sum_{e\in[\rho]:b_e\neq a_e^\star} \tilde{\lambda}_{e,b_e}^{-1}}{\tilde{\lambda}_{e,a_e}\Delta(a)^2}$, is such that $\tilde{v} \in \mathcal{F}(\theta)$. This, in turn, implies that

$$(14) = \inf_{\tilde{\lambda}\in\tilde{\Lambda}} \max_{a\neq a_\theta^\star} \frac{\sum_{e\in[\rho],a_e\in\mathcal{A}_e} \tilde{\lambda}_{e,a_e}(\theta_e(a_e^\star) - \theta_e(a_e))}{\frac{\sum_{e\in[\rho]:b_e\neq a_e^\star} \tilde{\lambda}_{e,b_e}^{-1}}{\Delta(a)^2}} \ge \inf_{\tilde{v}\in\mathcal{F}(\theta)} \sum_{e\in[\rho],a_e\in\mathcal{A}_e} \tilde{v}_{e,a_e}(\theta_e(a_e^\star) - \theta_e(a_e)).$$

On the other hand, we have that if $\tilde{v} \in \mathcal{F}(\theta)$, any $\tilde{\lambda}$, such that for all $a \neq a_\theta^\star$

$$\tilde{\lambda}_{e,a_e} = \frac{\tilde{v}_{e,a_e}}{\sum_{e\in[\rho]:b_e\neq a_e^\star} \tilde{v}_{e,a_e}}$$

is such that $\tilde{v} \in \mathcal{V}$, which in turn, yields

$$\inf_{\tilde{v}\in\mathcal{F}(\theta)} \sum_{e\in[\rho],a_e\in\mathcal{A}_e} \tilde{v}_{e,a_e}(\theta_e(a_e^\star) - \theta_e(a_e)) \ge \inf_{\tilde{v}\in\mathcal{F}(\theta)} \frac{\sum_{e\in[\rho],b_e\in\mathcal{A}_e} \tilde{\lambda}_{e,a_e}(\theta_e(a_e^\star) - \theta_e(a_e))}{\frac{\sum_{e\in[\rho]:a_e\neq a_e^\star} \tilde{\lambda}_{e,b_e}^{-1}}{\Delta(a)^2}} = (14)$$

$\square$

# B Non-convexity of the classical mean-field approximation

In this appendix, we establish that leveraging a classical mean-field approximation for $C_\theta^{\psi\text{-MF}}$ leads to a non-convex program. The conventional MF approximation [39] consists in selecting $\psi(w) = \prod_{i \in [N]} w_{i,a_i}$, for all $a \neq a^\star$. This choice results in a Signomial Program (SP) for $C_\theta^{\psi\text{-MF}}$, a specific type of non-convex optimization problem that is hard to solve efficiently [5]. We summarize this fact in the following lemma.

**Lemma B.1.** *Let* $\psi(w) = \prod_{i \in [N]} w_{i,a_i}, \forall a \neq a^\star$. *Then* $C_\theta^{\psi\text{-MF}}$ *is the value of the following signomial program:*

$$\min_{w \in \mathbb{R}_{\geq 0}^{KN}} \sum_{e \in [\rho], a_e \in \mathcal{A}_e} \zeta_{e,a_e}(w)(\theta_e(a_e^\star) - \theta_e(a_e)) \quad \text{s.t.} \sum_{e \in [\rho]: a_e \neq a_e^\star} \zeta_{e,a_e}^{-1}(w) \leq \Delta(a)^2, \forall a \in \mathcal{A}, \quad (15)$$

*where* $\zeta_{e,a_e}(w) = \prod_{i \in \mathcal{S}_e} w_{i,a_i} \prod_{j \notin \mathcal{S}_e} \left( \sum_{a_j \in \mathcal{A}_j} w_{j,a_j} \right), \forall e \in [\rho], a_e \in \mathcal{A}_e$.

*Proof.* Recall that, an SP in standard form (for variables $x \in \mathbb{R}_{>0}^I$) is described as [5, Sec. 9]:

$$\begin{aligned} \inf_x \quad & f(x) \\ \text{s.t.} \quad & g_i(x) \leq 1, \quad i = 1, \dots, m \\ & h_i(x) = 1, \quad i = 1, \dots, p \end{aligned} \tag{16}$$

where $f, (g_i)_{i \in [m]}$ and $(h_i)_{i \in [p]}$ are *signomials*. Recall that for $x \in \mathbb{R}_{>0}^I$, a signomial $F$ is a function of the type $F(x) = \sum_{j \in [J]} \xi_j \prod_{i \in [I]} x_i^{\beta_{i,j}}$, for some $\beta \in \mathbb{R}^{I \times J}$ and $\xi \in \mathbb{R}^J$.

Now, we will show that (16) can be described as an SP in standard form in terms of local variables $w \in \mathbb{R}_{>0}^{KN}$. By definition any $\tilde{v} \in \tilde{\mathcal{V}}_{\psi\text{-MF}}$ can be written as:

$$\tilde{v}_{e,a_e} = \sum_{b \in \mathcal{A}: b_e = a_e} w_b = \sum_{b \in \mathcal{A}: b_e = a_e} \prod_{i \in [N]} w_{i,b_i} = \prod_{i \in \mathcal{S}_e} w_{i,b_i} \prod_{j \notin \mathcal{S}_e} \left( \sum_{a_j \in \mathcal{A}_j} w_{j,a_j} \right) = \zeta_{e,a_e}(w) \quad (17)$$

The RHS of (17) is clearly a posynomial, in the local variables $w \in \mathbb{R}_{>0}^{KN}$. Also, as each $\tilde{v}_{e,a_e}$ is uniquely determined by $w$, we denote (17) by $\zeta_{e,a_e}(w)$, and define $\zeta(w) = (\zeta_{e,a_e}(w))_{e \in [\rho], a_e \in \mathcal{A}_e}$. Now, the objective of (15) is naturally a signomial in $w$ (note that $\theta_e(a_e^\star) - \theta_e(a_e)$ can be negative), as signomials are closed under addition and multiplication. The inequality constraints (15) can be written also in terms of $\zeta(w)$ and yields a posynomial in $w$. $\qquad\square$

*Remark* B.2 (Impossibility of reduction to GP). SPs are a generalization of GPs which are much harder to solve. Indeed GPs can be transformed into convex optimization programs, while only local solutions can be found efficiently for SPs [5]. Boyd et al. [5, Sec. 9.1] presents a set of conditions under which this transformation is possible. Specifically, the SP is first transformed into a Geometric Program (GP) through a set of relaxations and change of variables, which in turn is equivalent to a convex optimization problem. Unfortunately, for (15), it is easy to check that these conditions do not hold when selecting $\psi(w) = \prod_{i \in [N]} w_{i,a_i}$ in the $\psi$-MF approximation, which is a natural choice in approximate inference methods [39]a.

# C Technical lemmas

In this appendix, we state a set of technical lemmas useful to prove the regret upper bound. Specifically, in Lemma C.1, we state and adaptation of a concentration result first proposed in Lipschitz bandits [28] (Theorem 2) and then repurposed for combinatorial bandits in [10] (Lemma 1) and for general structured bandits in [8].

**Lemma C.1.** *There exists a constant $M(\gamma, \rho) > 0$ depending only on $\rho$ and $\gamma$ such that, for any $\theta$, and for all $t \geq 2$ satisfies*

$$\sum_{t=1}^{\infty} \mathbb{P}\left[\sum_{e\in[\rho], a_e \in \mathcal{A}_e} N_{t,e,a_e} \mathrm{kl}(\hat{\theta}_{t,e,a_e}, \theta_e(a_e)) \geq (1+\gamma)\log(t)\right] \leq M(\gamma, \rho),$$

*where $\mathrm{kl}(a, b)$ denotes the $\mathrm{kl}$ divergence between two Gaussian distributions with means $a$ and $b$.*

Next, in Lemma C.2 we present a result that allows to bound the size of a set of rounds in which a group action $a_e$ is selected and a group parameter $\theta_e(a_e)$ is not estimated accurately. It was first proposed for unimodal bandits [9] but is a result holding for general structured bandit problems [8].

**Lemma C.2.** *Let $a \in \mathcal{A}$ and $\varepsilon > 0$. Let $\mathcal{F}_t$ be the $\sigma$-algebra generated by $(r_{s,a})_{s\in[t]}$. Let $\mathcal{T} \subset \mathbb{N}$ be a random set of rounds. Assume that there exists a sequence of (random) sets $(\mathcal{T}(s))_{s \geq 1}$ such that (i) $\mathcal{T} \subset \cup_{s\in[t]}\mathcal{T}(s)$, (ii) for all $s \geq 1$ and $t \in \mathcal{T}(s)$, $N_{t,e,a_e} \geq \varepsilon s$, $\forall e \in [\rho]$ (iii) $|\mathcal{T}(s)| \leq 1$, and (iv) the event $t \in \mathcal{T}(s)$ is $\mathcal{F}_t$-measurable. Then, for all $\delta > 0$,*

$$\sum_{t=1}^{\infty} \mathbb{P}\left[t \in \mathcal{T}, \left|\sum_{e\in[\rho]} \hat{\theta}_{t,e,a_e} - \theta_e(a_e)\right| > \delta\right] \leq \frac{1}{\varepsilon\delta^2}.$$

Finally, in Lemma C.3, we report a result on the continuity of the functions related to the lower bound optimization problems and approximations. The result was first derived for the general structured bandit problem [8] and holds for MAMABs. Define $\tilde{v}^\diamond(m, \theta)$, the variables attaining $C_\theta^\diamond(m)$ for any $\theta$ and $m \in [K^N - 1]$.

In order to simplify the proofs, and w.l.o.g., in the following we shall assume that $\tilde{v}^\diamond(m, \theta)$ is unique, i.e., the problem (1) admits a unique solution. Note that if this was not the case, one may reason in terms of the objective function as in [22] for linear bandits. Furthermore, to simplify notations, we will omit the dependency of $m$ from $\tilde{v}^\diamond(m, \theta)$ and $C_\theta^\diamond(m)$, implying that the result holds for any $m \in [K^N - 1]$.

**Lemma C.3.** *Let $\diamond = \{L, MF\}$. The optimal value of (1), $\theta \to C_\theta^\diamond$ is continuous. If $C_\theta^\diamond$ admits a unique solution, $\tilde{v}^\diamond(\theta) = (\tilde{v}_{e,a_e}^\diamond(\theta))_{e\in[\rho], a_e\in\mathcal{A}_e}$ at $\theta$, then $\theta \to \tilde{v}^\diamond(\theta)$ is continuous at $\theta$.*

# D Regret upper bound

The proof for the regret upper bound consists in bounding the number of times a sub-optimal global action $a \neq a_\theta^\star$ is selected in the various phases of the algorithm. It is similar to the corresponding upper bound proof for OSSB [10, Th. 1]. The main difference is that the exploitation condition is based on group variables $\tilde{v}^\diamond \in \tilde{\mathcal{V}}_\diamond$, instead of global variables $v \in \mathbb{R}_{\geq 0}^A$.

For a parameter $\theta$, and $\diamond \in \{L, MF\}$ let $\tilde{v}^\diamond(\theta) = (\tilde{v}_{e,a_e}^\diamond)_{e\in[\rho], a_e\in\mathcal{A}_e}$ be the group variable vector solution of $C_\theta^\diamond$. We also denote by $\tilde{v}_t^\diamond = (\tilde{v}_{t,e,a_e}^\diamond)_{e\in[\rho], a_e\in\mathcal{A}_e}$ the group variable vector solution of $C_{\hat{\theta}_t}^\diamond$. Define $\psi^\diamond(\theta) = \tilde{A}\|\tilde{v}^\diamond(\theta)\|_\infty \sum_{e\in[\rho], a_e\in\mathcal{A}_e}(\theta_e(a_e^\star) - \theta_e(a_e))$. In this section, we use the following definition for the $\infty$ norm for the parameter $\theta$: $\|\theta\|_\infty = \max_{e\in[\rho], a_e\in\mathcal{A}_e}|\theta_e(a_e)|$. Let $(\kappa, \delta(\kappa))$ be such that, for all $\lambda \in \Theta$ verifying $\|\theta - \lambda\|_\infty \leq \delta(\kappa)$, the following holds:

$$C_\lambda^\diamond \leq (1+\kappa)C_\theta^\diamond,$$
$$\psi^\diamond(\lambda) \leq 2\psi^\diamond(\theta),$$
$$a_\theta^\star = a_\lambda^\star.$$

By Lemma C.3 such $\delta(\kappa)$ exist.

For any $\theta$ and $\chi > 0$, define

$$\Gamma(\theta, \chi) = \{\lambda \in \Theta : |\theta(a_\theta^\star) - \theta(a_\lambda^\star)| \leq \chi, a_\theta^\star \neq a_\lambda^\star\}. \tag{18}$$

To prove the upper bound, we further need to state the following assumption:

**Assumption D.1.** For any $\theta$, $\exists \chi(\theta) > 0$ s.t. if $\lambda \in \Gamma(\theta, \chi(\theta))$, there exists a parameter $\pi \in B(\lambda)$, $\forall e \in [\rho], \forall a_e \in \mathcal{A}_e$ such that

$$\mathrm{kl}(\theta_e(a_e), \lambda_e(a_e)) \geq \mathrm{kl}(\pi_e(a_e), \lambda_e(a_e)) - \frac{1}{2M^\diamond \tilde{A}},$$

where $M^\diamond$ is an upper bound of $\sup_{\theta \in \Theta} \|\tilde{v}^\diamond(\theta)\|_\infty$.

**Exploitation.** In this phase, ESM selects the best estimated global action $a_t = a_{\hat{\theta}_t}^\star$. Let $a \neq a_\theta^\star$ be a sub-optimal action, and define the event:

$$\mathcal{E}_a(t) = \left\{ a_t = a_{\hat{\theta}_t}^\star = a, N_{t,e,a_e} \mathrm{kl}(\hat{\theta}_{t,e,a_e}, \theta_e(a_e)) \geq (1 + \gamma) \log(t), \forall e \in [\rho], a_e \in \mathcal{A}_e \right\}.$$

Let $\chi > 0$ be a constant satisfying Assumption D.1. We decompose this event as

$$\mathcal{E}_{a,1}(t) = \mathcal{E}_a(t) \cap \left\{ \left| \sum_{e \in [\rho]} \theta_e(a_e) - \hat{\theta}_{t,e,a_e} \right| > \chi \right\},$$

$$\mathcal{E}_{a,2}(t) = \mathcal{E}_a(t) \cap \left\{ \left| \sum_{e \in [\rho]} \theta_e(a_e) - \hat{\theta}_{t,e,a_e} \right| \leq \chi \right\}.$$

Applying Lemma C.2, we have that

$$\sum_{t=1}^{\infty} \mathbb{P}(\mathcal{E}_{a,1}(t)) \leq \frac{1}{\chi^2}. \tag{19}$$

Now, assume that $\mathcal{E}_{a,2}(t)$ holds. We can write:

$$\sum_{e \in [\rho], a_e \in \mathcal{A}_e} N_{t,e,a_e} \mathrm{kl}(\hat{\theta}_{t,e,a_e}, \theta_e(a_e))$$

$$\overset{(i)}{\geq} (1 + \gamma) \log(t) \sum_{e \in [\rho], a_e \in \mathcal{A}_e} \tilde{v}_{t,e,a_e}^\diamond \mathrm{kl}(\hat{\theta}_{t,e,a_e}, \theta_e(a_e))$$

$$\overset{(ii)}{\geq} (1 + \gamma) \log(t) \sum_{e \in [\rho], a_e \in \mathcal{A}_e} \tilde{v}_{t,e,a_e}^\diamond \left( \mathrm{kl}(\hat{\theta}_{t,e,a_e}, \pi_e(a_e)) - \frac{1}{2M^\diamond \tilde{A}} \right)$$

$$\overset{(iii)}{\geq} (1 + \gamma) \log(t) \left( 1 - \sum_{e \in [\rho], a_e \in \mathcal{A}_e} \tilde{v}_{t,e,a_e}^\diamond \frac{1}{2M^\diamond \tilde{A}} \right)$$

$$\geq (1 + \gamma) \frac{\log(t)}{2},$$

where $(i)$ holds as we are in the exploitation phase, and hence the condition $N_{t,e,a_e} \geq \tilde{v}_{t,e,a_e}^\diamond (1 + \gamma) \log(t)$, holds for all $e \in [\rho], a_e \in \mathcal{A}_e$; $(ii)$ follows from Assumption D.1 which states that there exists a parameter $\pi \in B(\hat{\theta}_t)$ such that $\mathrm{kl}(\theta_e(a_e), \lambda_e(a_e)) \geq \mathrm{kl}(\pi_e(a_e), \lambda_e(a_e)) - \frac{1}{2M^\diamond \tilde{A}}$; $(iii)$ follows as $\pi \in B(\hat{\theta}_t)$.

We have hence shown that

$$\mathcal{E}_{a,2}(t) \subset \left\{ \sum_{e, a_e} N_{t,e,a_e} \mathrm{kl}(\hat{\theta}_{t,e,a_e}, \theta_e(a_e)) \geq (1 + \gamma) \frac{\log(t)}{2} \right\}.$$

From Lemma C.1, we have that:

$$\sum_{t=1}^{\infty} \mathbb{P}(\mathcal{E}_{a,2}(t)) \le G(\gamma, \rho). \tag{20}$$

Let $\mathcal{E}(t) = \cup_{a \ne a^\star} \mathcal{E}_a(t)$. Then, combining (19), (20), we conclude that

$$\sum_{t=1}^{\infty} \mathbb{P}(\mathcal{E}(t)) \le |\mathcal{A}| \left( G(\gamma, \rho) + \frac{1}{\chi^2} \right). \tag{21}$$

**Certainty equivalence.** We now consider the case in which $\theta$ is not estimated accurately enough and a sub-optimal action is selected. Define the event:

$$\mathcal{F}(t) = \left\{ a_t \ne a_\theta^\star, \exists e \in [\rho], a_e \in \mathcal{A}_e : N_{t,e,a_e} < (1+\gamma)\tilde{v}_{t,e,a_e}^\diamond \log(t), \|\theta - \hat{\theta}_t\|_\infty > \delta(\kappa) \right\}.$$

$\mathcal{F}(t)$ is the event that a sub-optimal action is selected, that we are not in the exploitation phase, and that $\theta$ is not estimated accurately. Let $\mathcal{F}_{e,a_e}(t) = \mathcal{F}(t) \cap \{|\hat{\theta}_{t,e,a_e} - \theta_e(a_e)| > \delta(\kappa)\}$ so that $\mathcal{F}(t) = \bigcup_{e,a_e} \mathcal{F}_{e,a_e}(t)$. We now show by contradiction that, if $\mathcal{F}(t)$ occurs, then $\min_{e,a_e} N_{t,e,a_e} \ge \varepsilon s_t/2$, for $\varepsilon < 1/|\mathcal{A}_0|$, where we recall that $s_t$ is the number of times in which ESM is not in the exploitation phase up to round $t$.

Assume that this does not hold, then there exist at least $p = \lceil s(t)/2 \rceil$ rounds $\{t_1, \dots, t_p\}$ where $\min_{e,a_e} N_{t_i,e,a_e} \le \varepsilon s_t$, for all $i \in [p]$. After $|\mathcal{A}_0|$ such rounds $\min_{e,a_e} N_{t,e,a_e}$ is increased of at least 1. This implies that $N_{t,e,a_e} \ge \frac{s_t}{2|\mathcal{A}_0|}$, but this is a contradiction for $\varepsilon < 1/|\mathcal{A}_0|$. Therefore, if $\mathcal{F}(t)$ occurs, then we have both $\min_{e\in[\rho],a_e\in\mathcal{A}_e} N_{t,e,a_e} \ge \frac{\varepsilon s_t}{2}$ and $\|\theta - \hat{\theta}_t\|_\infty > \delta(\kappa)$. By using a union bound and Lemma C.2, we have that

$$\sum_{t=1}^{\infty} \mathbb{P}(\mathcal{F}(t)) \le \sum_{t=1}^{\infty} \sum_{e\in[\rho],a_e\in\mathcal{A}_e} \mathbb{P}(\mathcal{F}_{e,a_e}(t)) \le \frac{2\tilde{A}}{\varepsilon \delta(\kappa)^2}. \tag{22}$$

**Estimation and exploration.** Now we consider the case in which $\theta$ is accurately estimated and we are not in the exploitation phase. Let

$$\mathcal{G}(t) = \left\{ \exists e \in [\rho], a_e \in \mathcal{A}_e : N_{t,e,a_e} < (1+\gamma)\tilde{v}_{t,e,a_e}^\diamond \log(t), \|\theta - \hat{\theta}_t\|_\infty \le \delta(\kappa) \right\}.$$

Hence, if $\mathcal{G}(t)$ occurs, then we know that ESM selects action according to the estimation or exploration phase. Assume that $\mathcal{G}(t)$ occurs and that $a_t = a$.

We first upper bound $s_t$ for these two cases.

(i) **Estimation:** In this case

$$a_t = a \in \mathcal{A}_0 : a_{t,e'} = b_{e'} \text{ with } (e', b_{e'}) \in \operatorname*{arg\,min}_{e,a_e} N_{t,e,a_e}.$$

Then we have that $N_{t,e,a_e} \le \min_{e,b_e} N_{t,e,b_e}$. Furthermore, since ESM is not in the exploitation phase, we have that there exist $e', a_e'$ such that

$$N_{t,e',a_e'} \le \tilde{v}_{t,e',a_e'}^\diamond (1+\gamma) \log(t) \le \|\tilde{v}_t^\diamond\|_\infty (1+\gamma) \log(T).$$

Hence we have that $N_{t,e,a_e} \le \|\tilde{v}_t^\diamond\|_\infty (1+\gamma) \log(T)$.

(ii) **Exploration:** In this case

$$a_t \in \mathcal{A} : a_{t,e'} = b_{e'} \text{ with } (e', b_{e'}) \in \operatorname*{arg\,min}_{e,a_e} \frac{N_{t,e,a_e}}{\tilde{v}_{t,e,a_e}}.$$

Then $N_{t,e,a_e} \le \tilde{v}_{t,e,a_e}^\diamond (1+\gamma) \log(t) \le \|\tilde{v}_t^\diamond\|_\infty (1+\gamma) \log(T)$.

Hence in both cases we get $N_{t,e,a_e} \le \|\tilde{v}_t^\diamond\|_\infty (1+\gamma)\log(t)$. Now, since $s_t$ is incremented each time $\mathcal{G}(t)$ occurs, we conclude that $s_t \le \tilde{A}\|\tilde{v}_t^\diamond\|_\infty(1+\gamma)\log(T)$.

We can now bound the number of times $a_e$ is selected in both phases. If $a_e$ is selected in the estimation phase, we have:
$$N_{t,e,a_e} \le \varepsilon s_t \le \varepsilon \tilde{A}\|\tilde{v}_t^\diamond\|_\infty(1+\gamma)\log(t).$$

Instead, when $a_e$ is selected in the exploration phase, we have that:
$$N_{t,e,a_e} \le \tilde{v}_{t,e,a_e}^\diamond (1+\gamma)\log(T),$$

We deduce that

$$\sum_{t=1}^T \mathbb{1}_{\{a_{t,e}=a_e,\mathcal{G}(t)\}} \le (\tilde{v}_{\tau,e,a_e}^\diamond + \varepsilon\tilde{A}\|\tilde{v}_\tau^\diamond\|_\infty)(1+\gamma)\log(T),$$

where $\tau \le T$ is the last random round such that $a_{\tau,e} = a_e$ and $\mathcal{G}(\tau)$ holds. The fact $\mathcal{G}(\tau)$ occurs, implies that $\|\hat\theta_\tau - \theta\|_\infty \le \delta(\kappa)$ holds, and we have

$$\sum_{e\in[\rho],a_e\in\mathcal{A}_e} \tilde{v}_{\tau,e,a_e}^\diamond (\theta_e(a_e^\star) - \theta_e(a_e)) \le (1+\kappa)C_\theta^\diamond,$$

$$\varepsilon|\mathcal{A}_0|\|\tilde{v}_\tau^\diamond\|_\infty \sum_{e\in[\rho],a_e\in\mathcal{A}_e} (\theta_e(a_e^\star) - \theta_e(a_e)) \le 2\varepsilon\psi^\diamond(\theta),$$

This event yield regret:

$$Y(T) = \sum_{t=1}^T \sum_{e\in[\rho],a_e\in\mathcal{A}_e} (\theta_e(a_e^\star) - \theta_e(a_e))\mathbb{1}_{\{a_{t,e}=a_e,\mathcal{G}(t)\}}.$$

Hence, we conclude that

$$Y(T) \le ((1+\kappa)C_\theta^\diamond + 2\varepsilon\psi^\diamond(\theta))(1+\gamma)\log(T). \tag{23}$$

**Regret upper bound.** By combining (21)-(22)-(23), we have:

$$R^\pi(T) \le \mathbb{E}[Y(T)] + \theta(a^\star)\left(\sum_{t=1}^\infty \mathbb{P}(\mathcal{E}(t)) + \mathbb{P}(\mathcal{F}(t))\right)$$

$$\le ((1+\kappa)C_\theta^\diamond + 2\varepsilon\psi^\diamond(\theta)(1+\gamma))\log(T) + \theta(a^\star)\left(|\mathcal{A}|\left(G(\gamma,\rho) + \frac{1}{\chi^2}\right) + \frac{2\tilde{A}}{\varepsilon\delta(\kappa)^2}\right),$$

which in turn implies

$$\limsup_{T\to\infty} \frac{R^\pi(T)}{\log(T)} \le ((1+\kappa)C_\theta^\diamond + 2\varepsilon\psi^\diamond(\theta))(1+\gamma).$$

As, the above holds for all $\kappa > 0$ we get the result:

$$\limsup_{T\to\infty} \frac{R^\pi(T)}{\log(T)} \le (C_\theta^\diamond + 2\varepsilon\psi^\diamond(\theta))(1+\gamma).$$

# E   Connection to Combinatorial Semi-bandit Feedback Bandits

The MAMAB setting can be regarded as a specific instance of a combinatorial semi-bandit feedback setting [13, 12, 38]. In the following, we present an equivalent characterization of the MAMAB problem to clarify its connection to the combinatorial semi-bandit feedback setting.

We first describe the interaction model in the generic (linear) combinatorial semi-bandit feedback setting. In such a setting, at each round $t \geq 1$, the learner selects an action from a combinatorial set $a_t \in \{0,1\}^d$, and, given an unknown parameter $\tilde{\theta} \in \mathbb{R}^d$, she observes:

$$r_{t,i} = \tilde{\theta}_i + \eta_{t,i}, \forall i \in [d] : a_{t,i} = 1,$$

where $\eta_{t,i} \sim \mathcal{N}(0,1)$, for all $i \in [d]$, are i.i.d. Gaussian noise samples (over rounds).

Since in MAMAB the set of global actions is defined as $\mathcal{A} = \times_{i \in [N]} \mathcal{A}_i$, the problem is not directly interpretable in the semi-bandit feedback setting. We show that a simple map from actions in $\mathcal{A}$ to binary vectors in the $\tilde{A}$-dimensional space can reduce the MAMAB problem to a problem in the semi-bandit feedback setting.

Let $\phi(\cdot) : \mathcal{A} \to \{0,1\}^{\tilde{A}}$ be a function mapping global actions to binary vectors in the $\tilde{A}$-dimensional space. In MAMAB, the vector $\phi$ has a block structure: it can be decomposed as $\phi(a) = (\phi_e(b_e))_{e \in [\rho], b_e \in \mathcal{A}_e}$, where $\phi_e(b_e) \in \{0,1\}^{A_e}$ is a group vector $\phi_e(b_e) = 1_{\{a_e = b_e\}}$, i.e., containing 1 in correspondence of the activated group action $a_e$. Further define $\tilde{\theta} = (\theta_e(a_e))_{e \in [\rho], a_e \in \mathcal{A}_e} \in \mathbb{R}^{\tilde{A}}$, i.e., $\tilde{\theta}$ is the vector containing the local mean parameters. At round $t \geq 1$, a global action $a_t \in \mathcal{A}$ is selected by the learner, and she observes:

$$r_{t,e,a_e} = \tilde{\theta}_e(a_e), \forall e \in [\rho] : \phi_e(a_{t,e}) = 1.$$

In other words, in the semi-bandit feedback setting, a (global) action $a \in \mathcal{A}$ is selected and the learner observes a vector of rewards $[r_e(a_e)]_{e \in [\rho], a_e \subseteq a}$, where $r_e(a_e) = \theta_e(a_e)^\top \phi_e(a_e) + \eta_e$, where $\eta_e \sim \mathcal{N}(0,1)$ is i.i.d. Gaussian Noise. Note that the feature vectors satisfy $\|\phi(a)\|_0 = \rho$, $\forall a \in \mathcal{A}$ and $\|\phi_e(a_e)\|_0 = 1$, $\forall e \in [\rho], a_e \in \mathcal{A}_e$. In order to further clarify the connection to the semi-bandit feedback, we provide a concrete example below.

*Example* 1. Consider the line factor graph in Fig. 7 with $N = 3$ agents, $\rho = 2$ groups, and $K = 2$ actions. The reward can be written as $r(a_1, a_2, a_3) = r_1(a_1, a_2) + r_2(a_2, a_3)$. Let $a_i \in \{0,1\}$, for all $i \in [N]$. The average reward can be expressed by the vector $\tilde{\theta} = ((\theta_1(a_1, a_2))_{(a_1, a_2) \in \{0,1\}^2}, (\theta_2(a_2, a_3))_{(a_2, a_3) \in \{0,1\}^2}) \in \mathbb{R}^8$, where

$$(\theta_1(a_1, a_2))_{(a_1, a_2) \in \{0,1\}^2} = (\theta_1(0,0), \theta_1(0,1), \theta_1(1,0), \theta_1(1,1))$$
$$(\theta_2(a_2, a_3))_{(a_2, a_3) \in \{0,1\}^2} = (\theta_2(0,0), \theta_2(0,1), \theta_2(1,0), \theta_2(1,1)).$$

For example, selecting action $a = (0,0,0)$, corresponds to the feature vector $\phi(a) = (1,0,0,0,1,0,0,0)$, while selecting action $b = (0,1,0)$ corresponds to the feature vector $\phi(b) = (0,1,0,0,0,0,1,0)$.

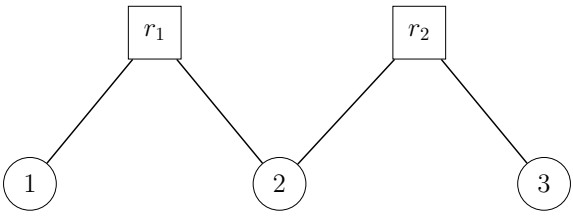

Figure 7: Factor graph from Example 1.

# F  Constraint reduction

## F.1  Proof of Lemma 5.4

*Proof.* First, we shall prove $C_\theta^\diamond(m+1) \leq C_\theta^\diamond(m), \forall m \in [K^N - 1]$, by induction. The base case, for $m = 1$, is $C_\theta^\diamond(2) \leq C_\theta^\diamond(1)$. Note that $C_\theta^\diamond(1)$ can be written as

$$\inf_{\tilde{v} \in \tilde{\mathcal{V}}_\diamond} \sum_{e \in [\rho], a_e \in \mathcal{A}_e} \tilde{v}_{a_e}(\theta_e(a_e^\star) - \theta_e(a_e)) \tag{24}$$

$$\text{s.t.} \sum_{e \in [\rho]: a_e^{(2)} \neq a_e^\star} \tilde{v}_{a_e^{(2)}}^{-1} \leq \Delta_1^2 \tag{25}$$

$$\sum_{e \in [\rho]: a_e \neq a_e^\star} \tilde{v}_{e,a_e}^{-1} \leq \Delta_1^2, \forall a \in \mathcal{A} \setminus \{a^{(1)}, a^{(2)}\}, \tag{26}$$

while $C_\theta^\diamond(2)$ is defined as

$$\inf_{\tilde{v} \in \tilde{\mathcal{V}}_\diamond} \sum_{e \in [\rho], a_e \in \mathcal{A}_e} \tilde{v}_{a_e}(\theta_e(a_e^\star) - \theta_e(a_e)) \tag{27}$$

$$\text{s.t.} \sum_{e \in [\rho]: a_e^{(2)} \neq a_e^\star} \tilde{v}_{a_e^{(2)}}^{-1} \leq \Delta_1^2 \tag{28}$$

$$\sum_{e \in [\rho]: a_e^{(3)} \neq a_e^\star} \tilde{v}_{a_e^{(3)}}^{-1} \leq \Delta_2^2 \tag{29}$$

$$\sum_{e \in [\rho]: a_e \neq a_e^\star} \tilde{v}_{e,a_e}^{-1} \leq \Delta_2^2, \forall a \in \mathcal{A} \setminus \{a^{(1)}, a^{(2)}, a^{(3)}\}. \tag{30}$$

Now, the constraints (25) and (28) are identical. The constraints (29)-(30) can be simply written as

$$\sum_{e \in [\rho]: a_e \neq a_e^\star} \tilde{v}_{e,a_e}^{-1} \leq \Delta_2^2, \forall a \in \mathcal{A} \setminus \{a^{(1)}, a^{(2)}\}.$$

The expression is identical to (26) with the exception of the term $\frac{1}{\Delta_1^2}$ in place of $\frac{1}{\Delta_2^2}$. As $\Delta_1 \leq \Delta_2$, we naturally conclude that $C_\theta^\diamond(2) \leq C_\theta^\diamond(1)$.

Now, assume that $C_\theta^\diamond(m'+1) \leq C_\theta^\diamond(m')$ holds for $m' = m - 1$, to complete the induction we need to show that $C_\theta^\diamond(m'+1) \leq C_\theta^\diamond(m')$, for $m' = m$. By following a similar approach to the base case, we can show that the only difference in the optimization problems defining $C_\theta^\diamond(m')$ and $C_\theta^\diamond(m'+1)$ is in the last set of constraints: for $C_\theta^\diamond(m')$ these constraints are

$$\sum_{e \in [\rho]: a_e \neq a_e^\star} \tilde{v}_{e,a_e}^{-1} \leq \Delta_{m'}^2, \forall a \in \mathcal{A} \setminus \cup_{j \in [m']} \{a^{(j+1)}\},$$

while for $C_\theta^\diamond(m+1)$ they can be written as

$$\sum_{e \in [\rho]: a_e \neq a_e^\star} \tilde{v}_{e,a_e}^{-1} \leq \Delta_{m'+1}^2, \forall a \in \mathcal{A} \setminus \cup_{j \in [m']} \{a^{(j+1)}\}.$$

Hence, we can directly conclude that $C_\theta^\diamond(m'+1) \leq C_\theta^\diamond(m')$, as it holds that $\Delta_m' \leq \Delta_{m'+1}$

Now, to complete the proof, we show that $C_\theta^\star \leq C_\theta^{\text{MF}}(m), \forall m \in [K^N - 1]$. Since we have that $C_\theta^{\text{MF}}(m+1) \leq C_\theta^{\text{MF}}(m)$, it is sufficient to prove that $C_\theta^\star \leq C_\theta^{\text{MF}}(K^N - 1)$. It is easy to check that $C_\theta^{\text{MF}}(K^N - 1)$ can be written as

$$\inf_{\tilde{v} \in \tilde{\mathcal{V}}_{\text{MF}}} \sum_{e \in [\rho], a_e \in \mathcal{A}_e} \tilde{v}_{a_e}(\theta_e(a_e^\star) - \theta_e(a_e)) \quad \text{s.t.} \sum_{e \in [\rho]: a_e \neq a_e^\star} \tilde{v}_{a_e}^{-1} \leq \Delta(a)^2, \forall a \neq a^\star. \tag{31}$$

Eq. (31) corresponds to $C_\theta^\star$ in (1), with the difference that the variables are in $\tilde{\mathcal{V}}_{\text{MF}}$. As $\tilde{\mathcal{V}}_{\text{MF}} \subseteq \tilde{\mathcal{V}}$ we conclude that $C_\theta^\star \leq C_\theta^{\text{MF}}(K^N - 1)$. $\qquad\square$

## G  Variable Elimination and Factored Constraint Reduction

First, we present VE (Alg. 2) and FCR (Alg. 3), two important sub-routines used in this paper. We use VE to select the best global action in the exploitation phase of ESM (Sec. 6), and FCR to represent in a compact way the exponentially large constraint set of the lower bound problems 8. We then report known results on the complexity and correctness of these algorithms. To clarify their use, we present examples of the application of these methods on specific factor graphs to compute the global best arm. We finally present algorithms to efficiently compute the $m$-best global actions used in the constraint reduction $C_\theta^\diamond(m)$

### G.1  Variable Elimination

The VE algorithm [14] is a classical algorithm for probabilistic graphical models used for a variety of exact inference tasks (e.g., maximum a posteriori, computation of marginals, etc. [27]). It involves iteratively eliminating variables by combining and marginalizing factors to find the most probable assignment until only the query variables remain. Specifically, VE takes as input an elimination order $\mathcal{O}$, where $\mathcal{O}(i)$ is the $i^{th}$ variable to be eliminated and a set of factored functions $\mathcal{R} = \{r_e\}_{e \in [\rho]}$. Each factor $r_e$ is a function mapping $a_e \in \mathcal{A}_e$ to real values. For a factor $r_e$, we denote its scope by $\mathrm{Sc}(r_e) \subseteq [N]$, which represents the set of variables involved in the factor.

The algorithm proceeds iteratively for $i = 1, \ldots, N$, by eliminating variable $l = \mathcal{O}(i)$ in each round. In round $i$, all the factors $r_e$ containing variable $l$ in their scopes are collected in the set $\mathcal{R}_l$. Subsequently, the (marginal) best response for agent $l$ is computed as $p_l(a_{e \setminus l}) = \max_{a_l \in \mathcal{A}_l} \sum_{r_e \in \mathcal{R}_l} r_e(a_{e \setminus l}, a_l)$, where $a_{e \setminus l}$ corresponds to the action $a_e$ corresponds to the action $a_e$ with the component corresponding to the $l$-th agent is removed. The set of factors is then updated as $\mathcal{R} \leftarrow \mathcal{R} \cup \{p(a_{e \setminus l})\} \setminus \mathcal{R}_l$. At this point, every factor containing $l$ in its scope is eliminated. At the next iteration, the algorithm selects the next variable to be eliminated and repeats this procedure until $i = N$. Finally, it returns the optimal value $\sum_{p \in \mathcal{R}} p_{\mathcal{O}(N)}$.

---

**Algorithm 2** VE

**Input:** Elimination order $\mathcal{O}$, factors $\mathcal{R}$
**for** $i = 1, \ldots, N$ **do**
$\qquad l = \mathcal{O}(i)$
$\qquad \mathcal{R}_l = \{r_e \in \mathcal{R} : l \in \mathrm{Sc}(r_e)\}$
$\qquad p_l(a_{e \setminus l}) = \max_{a_l \in \mathcal{A}_l} \sum_{r_e \in \mathcal{R}_l} r_e(a_l, a_{e \setminus l})$
$\qquad \mathcal{R} \leftarrow \mathcal{R} \cup \{p_l(a_{e \setminus l})\} \setminus \mathcal{R}_l$
**return** $\sum_{p \in \mathcal{R}} p_{\mathcal{O}(N)}$

---

**Complexity of VE.** VE is guaranteed to return the optimal global arm in $O(NK^{A_\mathcal{O}+1})$ operations [27], where $A_\mathcal{O} = \max_{i \in [N]} |\mathrm{Sc}(p_{\mathcal{O}(i)})|$ is the size of the largest factor generated when using elimination order $\mathcal{O}$. The complexity of VE depends on the elimination order $\mathcal{O}$ and is linear in the maximum size of the scope of "best-response functions" introduced in the elimination process. App. G.4 discuss the scaling of elimination orders for typical factor graphs.

**Example of VE application.** We now illustrate the use of VE to compute a global optimal arm $a_\theta^\star$ in the following example. Note that VE, applied to $\mathcal{R} = \{\theta_e\}_{e \in [\rho]}$ returns the highest global expected reward $\theta(a^\star)$. A backward pass of the VE algorithm recovers the optimal arm $a^\star$ as shown in the following example.

*Example* 2. Consider the factor graph in Fig. 8 with $N = \rho = 4$. The average reward is described as:

$$\theta(a) = \theta_1(a_1, a_2) + \theta_2(a_2, a_4) + \theta_3(a_1, a_3) + \theta_4(a_3, a_4).$$

The key idea in VE is that, rather than summing all reward functions and then doing the maximization, we fix an ordering for the variables, and we maximize over variables one at a time, according to the elimination order $\mathcal{O}$. For example, let $\mathcal{O} = \{a_4, a_3, a_2, a_1\}$. Starting from $a_4$, we get

$$\max_{a \in \mathcal{A}} \theta(a) = \max_{a_1, a_2, a_3} \theta_1(a_1, a_2) + \theta_3(a_1, a_3) + \max_{a_4} \theta_2(a_2, a_4) + \theta_4(a_3, a_4).$$

Agent 4 can summarize its marginal best response when varying $(a_2, a_3)$ using a new factor $p_4(a_2, a_3) = \max_{a_4} \theta_2(a_2, a_4) + \theta_4(a_3, a_4)$, which represents the best response of agent 4 condi-

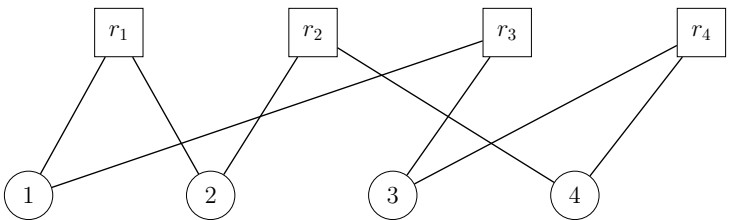

Figure 8: Factor graph from example 2.

tioned on the actions played by agents $2, 3$. We may also denote $a_4^\star(a_2, a_3) = \arg\max_{a_4} \theta_2(a_2, a_4) + \theta_4(a_3, a_4)$ as the best action for agent $4$ conditioned on the actions of agent $2, 3$. Hence, we get

$$\max_{a \in \mathcal{A}} \theta(a) = \max_{a_1, a_2, a_3} \theta_1(a_1, a_2) + \theta_3(a_1, a_3) + p_4(a_2, a_3).$$

Similarly, agent 3, performs its marginal best response $p_3(a_1, a_2) = \max_{a_3} \theta_3(a_1, a_3) + p_4(a_2, a_3)$, and marginal best action $a_3^\star(a_1, a_2) = \arg\max_{a_3} \theta_3(a_1, a_3) + p_4(a_2, a_3)$. The problem is further reduced to

$$\max_{a \in \mathcal{A}} \theta(a) = \max_{a_1, a_2} \theta_1(a_1, a_2) + p_3(a_1, a_2).$$

Next, agent 2 computes her marginal best response $p_2(a_1) = \max_{a_2} \theta_1(a_1, a_2) + p_3(a_1, a_2)$, and marginal best action $a_2^\star(a_1) = \arg\max_{a_2} \theta_1(a_1, a_2) + p_3(a_1, a_2)$.

Finally, agent 1 selects compute the best response $p_1 = \max_{a_1} p_2(a_1)$, and we have

$$\max_{a \in \mathcal{A}} \theta(a) = \max_{a_1} p_2(a_1).$$

We can recover the best global action $a^\star = (a_1^\star, a_2^\star, a_3^\star, a_4^\star)$ by performing the entire process in reverse order: $a_1^\star = \arg\max_{a_1} p_2(a_1)$, $a_2^\star = \arg\max_{a_2} \theta_1(a_1^\star, a_2) + p_3(a_1^\star, a_2)$, $a_3^\star = \arg\max_{a_3} \theta_3(a_1^\star, a_3) + p_4(a_2^\star, a_3)$, and $a_4^\star = \arg\max_{a_4} \theta_2(a_2^\star, a_4) + \theta_4(a_3^\star, a_4)$.

Note that the complexity of VE for this example is $O(K^3)$ floating point operations, while naively finding the best global action would require $O(K^4)$ operations.

### G.2 Factored Constraint Reduction

The FCR algorithm follows a similar idea to VE to represent a set of factorized constraints in a compact manner and is inspired by the Factored-LP algorithm [20] to reduce constraints in the Bellman LP for Factored MDPs. FCR considers constraints of the type:

$$\mathcal{C} = \left\{ \sum_{e \in [\rho]} p_e(a_e) \leq c, \forall a \in \mathcal{A} \right\},$$

where $p_e(\cdot)$ is a factor function mapping local actions $a_e \in \mathcal{A}_e$ to real values, and $c$ is a constant, and construct an equivalent set of constraints $\mathcal{K}$ of reduced size. We present the pseudo-code of FCR in Alg. 3 and describe its steps below.

FCR takes as input an initial set of factors $\mathcal{F} = \{p_e, \forall e \in [\rho]\}$, and an ordered elimination set $\mathcal{O}$. For a factor $p \in \mathcal{F}$, we define its *scope* $\mathrm{Sc}(p) \subseteq [N]$ as the set of agents involved in $p$. We also associate a real variable to each factor $p \in \mathcal{F}$, $u^p_{a_{\mathrm{Sc}(p)}}$. After initializing the output constraint set as $\mathcal{K} = \emptyset$, the algorithm proceeds in an iterative manner. At each iteration $i = 1, \ldots, N$, we set $l = \mathcal{O}(i)$ (the $i^{th}$ element of $\mathcal{O}$), and define $\mathcal{F}_l = \{p \in \mathcal{F} : l \in \mathrm{Sc}(p)\}$. We then introduce a new factor $p_l$ having scope $\mathrm{Sc}(p_l) = \cup_{p \in \mathcal{F}_l}\{\mathrm{Sc}(p)\} \setminus \{l\}$, and we associate the variable $u^{p_l}_{a_{\mathrm{Sc}(p_l)}}$ to $p_l$. We include in $\mathcal{K}$ a new set of constraints

$$u^{p_l}_{a_{\mathrm{Sc}(p_l)}} \geq \sum_{p \in \mathcal{F}_l} u^p_{a_{\mathrm{Sc}(p)}}, \quad \forall a_{\mathrm{Sc}(p_l)}, a_l.$$

We further include the new factor variable $p_l$ in the set of factors $\mathcal{F}$ and remove all factors in $\mathcal{F}_l$ from it, i.e., $\mathcal{F} = \mathcal{F} \cup \{p_l\} \setminus \mathcal{F}_l$. At $l = \mathcal{O}(N)$, we introduce the constraint $u^{p_{\mathcal{O}(N)}} \leq c$ into $\mathcal{K}$, where $p_{\mathcal{O}(N)}$ is the last generated factor and has empty scope.

---

**Algorithm 3** FCR

---

**Input:** Elimination order $\mathcal{O}$, factors $\mathcal{F}$
**Initialize** $\mathcal{K} = \emptyset$
**for** $i = 1, \ldots, N$ **do**
    $l \leftarrow \mathcal{O}(i)$
    $\mathcal{F}_l \leftarrow \{p \in \mathcal{F} : l \in \mathrm{Sc}(p)\}$
    $\mathcal{K} \leftarrow \mathcal{K} \cup \left\{ u^{p_l}_{a_{\mathrm{Sc}(p_l)}} \geq \sum_{p \in \mathcal{F}_l} u^{p}_{a_{\mathrm{Sc}(p)}}, \forall a_{\mathrm{Sc}(p_l)}, a_l \right\}$
    $\mathcal{F} \leftarrow \mathcal{F} \cup \{p_l\} \setminus \mathcal{F}_l$
  $\mathcal{K} \leftarrow \mathcal{K} \cup \{u^{p_{\mathcal{O}(N)}} \leq c\}$
**return** $\mathcal{K}$

---

**Complexity of FCR.** The properties of FCR are directly inherited by the ones of VE. First, FCR is guaranteed to return a provably equivalent representation of the set of constraints [19, Thm. 4.4]. Specifically, let $\mathcal{K} = \mathrm{FCR}(\mathcal{C})$. Then $\mathcal{C}$ and $\mathcal{K}$ are equivalent, that is, an assignment of variables $(u, \tilde{v})$ is feasible for $\mathcal{K}$ if and only if $\tilde{v}$ is feasible for $\mathcal{C}$. Furthermore, similarly to VE, the number of constraints and variables to represent $\mathcal{C}$ scales linearly in $N$ and exponentially in $A_{\mathcal{O}} = \max_{i \in [N]} |\mathrm{Sc}(p_{\mathcal{O}(i)})|$, i.e., the size of the largest scope when using the elimination order $\mathcal{O}$. Specifically, the number of constraints in $\mathcal{K}$ scales as $O(NK^{A_{\mathcal{O}}+1})$. Note that FCR also includes $O(NK^{A_{\mathcal{O}}})$ new (scalar) variables in the optimization problem. Note that the proof of Lemma 5.5 is a direct consequence of this result (see [19, Thm. 4.4] for details).

**Example of FCR application.** Let $m \in [K^N - 1]$. We provide an example of the application of FCR to reduce the combinatorial number of constraints appearing in $(8)^1$ :

$$\sum_{e \in [\rho] : a_e \neq a_e^\star} \tilde{v}^{-1}_{a_e} \leq \Delta^2_m, \ \forall a \in \mathcal{A}.$$

*Example* 3. Consider the factor graph in Ex. 2. Let $f_e(a_e) = \tilde{v}^{-1}_{a_e} \mathbb{1}_{\{a_e \neq a_e^\star\}}$, for all $e \in [\rho], a_e \in \mathcal{A}_e$. Then the set of constraints can be written as

$$\sum_{e \in [\rho]} f_e(a_e) \leq \Delta^2_m, \forall a \in \mathcal{A},$$

or equivalently as

$$\Delta^2_m \geq \max_{a_1, a_2, a_3, a_4} f_1(a_1, a_2) + f_2(a_2, a_4) + f_3(a_1, a_3) + f_4(a_3, a_4).$$

We introduce a set of variables $(u^{f_e}_{a_e})_{e \in [\rho], a_e \in \mathcal{A}_e}$, and the equality constraints:

$$u^{f_e}_{a_e} = \tilde{v}^{-1}_{a_e}, \forall e \in [\rho], a_e \in \mathcal{A}_e.$$

Note that we can rewrite $f_e(a_e) = u^{f_e}_{a_e}$. Then, we fix an elimination ordering $\mathcal{O} = \{4, 3, 2, 1\}$ and initialize $\mathcal{F} = \emptyset, \mathcal{K} = \emptyset$. Now we introduce a new "function" $p_l$ into $\mathcal{F}$ by eliminating a variable $l = \mathcal{O}(i)$, at each round $i = 1, \ldots, 4$.

For $i = 1$, we have $\mathcal{O}(1) = 4$ and $\mathcal{F}_1 = \{f_2(a_2, a_4), f_4(a_3, a_4)\}$. We define a new variable $u^{p_4}_{a_2, a_3}$ associated to $p_4$, and introduce the set of constraints:

$$u^{p_4}_{a_2, a_3} \geq u^{f_2}_{a_4, a_2} + u^{f_4}_{a_4, a_3}, \forall (a_2, a_3, a_4) \in \mathcal{A}_2 \times \mathcal{A}_3 \times \mathcal{A}_4.$$

These in constraints are included in the set $\mathcal{K}$. We further exclude the function $f_2$ and $f_4$ from the set $\mathcal{F}$, while including $p_4$ in it.

Subsequently, we consider $i = 2$ and $\mathcal{O}(2) = 3$. Then $\mathcal{F}_3 = \{p_4(a_2, a_3), f_3(a_3, a_1)\}$. We introduce the new set of constraints:

$$u^{p_3}_{a_1, a_2} \geq u^{p_4}_{a_2, a_3} + u^{f_3}_{a_3, a_1}, \forall (a_1, a_2, a_3) \in \mathcal{A}_1 \times \mathcal{A}_2 \times \mathcal{A}_3,$$

---

[1]Note that the original constraints (8) are defined for actions $a \in \mathcal{A} \setminus \cup_{j \in [m]} \{a^{(j+1)}\}$. However the redundant constraints $a \in \cup_{j \in [m]} \{a^{(j+1)}\}$ are inactive since the constraints $\sum_{e \in [\rho] : a_e^{(j+1)} \neq a_e^\star} \tilde{v}^{-1}_{e, a_e^{(j+1)}} \leq \Delta^2_j, \forall j \in [m]$ appearing in the problem defining $C_\theta^\diamond$ are tighter. Hence these constraints may be included w.l.o.g.

and we add them to the constraint set $\mathcal{K}$. We proceed to eliminate $p_4$ and $f_3$ from $\mathcal{F}$ and include $p_3$.

We then move to $i = 3$, $\mathcal{O}(3) = 2$ and define $\mathcal{F}_2 = \{f_1(a_1, a_2), p_3(a_1, a_2)\}$. The set of constraints introduced at this step are:

$$u_{a_1}^{p_2} \geq u_{a_1,a_2}^{p_3} + u_{a_1,a_2}^{f_1}, \forall(a_1, a_2) \in \mathcal{A}_1 \times \mathcal{A}_2,$$

and similarly to the previous steps we add these constraints to $\mathcal{K}$ and eliminate the variables $p_3$ and $f_1$ from $\mathcal{F}$, while including $p_2$. The last step at $\mathcal{O}(4) = 1$ consists of including in $\mathcal{K}$ the constraints

$$u^{p_1} \geq u_{a_1}^{p_2}, \forall a_1 \in \mathcal{A}_1.$$

Finally we add to $\mathcal{K}$ the constraint $u^{p_1} \leq \Delta_m^2$, and output $\mathcal{K}$. The number of constraints in the transformed set is $|\mathcal{K}| = 2K^3 + K^2 + K + 1$, while the original set has $|\mathcal{C}| = K^4 - 1$ constraints.

### G.3 $m$-BEST algorithm

In this section, we discuss an algorithm to find the $m$-best global arms. As explained in App. F, a set of tighter approximations $C_\theta^\diamond(m)$, for $m \in [K^N - 1]$, can be built by considering an ordering of the first $m$ smallest gaps and hence requires to compute the $m + 1$ global arms with highest expected rewards. The Lawler and Nilsson's $m$-BEST algorithm [26, 30], briefly described in the remainder of this section, will serve this purpose.

The procedure was originally devised to compute the $m$ most probable configurations in graphical models. The main idea is the following: At each step, the m-BEST find the best solution to a re-formulation of the original problem that excludes the solutions already discovered. Specifically, at each time iteration $j < m$, the algorithm runs VE excluding the first $j$ most probable configurations. The Lawler's algorithm [26] starts by computing the best global action $a^{(1)}$ by applying VE (with elimination order $\mathcal{O}$) over the combinatorial action space $\mathcal{A}$ by applying VE $N$ times. To determine the second best action $a^{(2)}$, the algorithm searches over the set $\mathcal{A}_{(2)} = \mathcal{A} \setminus \{a^{(1)}\}$. More generally, at iteration $j$, the algorithm finds the $j^{\text{th}}$ best global action $a^{(j)}$ by running VE over the sets $\mathcal{A}_{(j)} = \mathcal{A} \setminus \cup_{k \in [j]} \{a^{(k)}\}$. This procedure provably identifies the $m$-best global actions with complexity $O(mN^2 K^{A_\mathcal{O}+1})$. By leveraging similar ideas and using a junction tree representation of the graph, Nilsson [30] improves over this procedure leading to an $m$-best algorithm with complexity $O(mNK^{A_\mathcal{O}+1})$.

### G.4 Examples of optimal elimination orders

The following lemma presents a few examples of optimal elimination orderings, i.e. orderings $\mathcal{O}$ that minimize $A_\mathcal{O}$, for specific factor graph topologies.

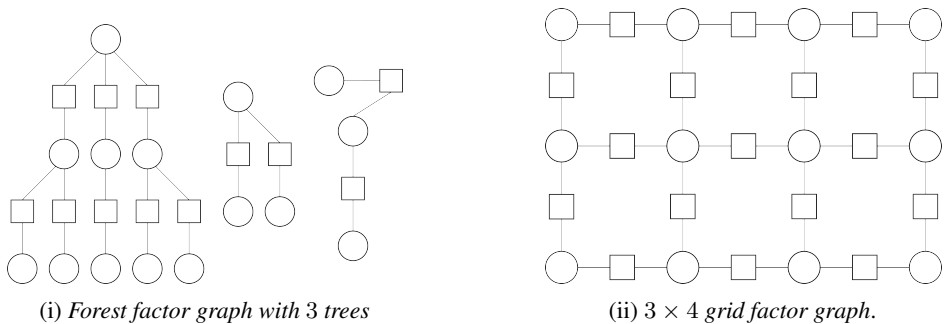

(i) *Forest factor graph with* 3 *trees*          (ii) $3 \times 4$ *grid factor graph.*

Figure 9: Examples of factor graphs.

**Lemma G.1.** *There exist known optimal orderings for the following factor graph structures. These orderings yield*

1. $A_\mathcal{O} = 2$ *for any forest factor graph,*

2. $A_\mathcal{O} = 1 + \min\{p, q\}$ *for any* $p \times q$ *grid factor graph.*

The lemma implies that there are optimal orderings for a *tree* (Fig. 9), *star*, and *line* (Fig. 3) factor graphs with $A_{\mathcal{O}} = 2$ (since are particular cases of forests), and for ring factor graphs 3 have $A_{\mathcal{O}} = 3$. It is easy to verify that the optimal elimination order for trees starts eliminating factors from the "leaves" of each tree composing the forest and proceeds upwards to the roots, while for grids the optimal ordering starts from any "corner" of the grid and proceeds inwards [39].

However, note that finding an optimal ordering for general graphs, is an $\mathcal{NP}$-hard problem [14], and the worst-case runtime of VE is exponential in $N$. This issue has been mitigated successfully for a large variety of graph structures in the graphical model community, where there exist a variety of heuristics for the VE ordering problem (see [27, Sec. 4.3.3] for a comprehensive overview of these methods).

## H Complexity results

We state a set of fundamental complexity results for various problems encountered in this paper.

**Lemma H.1.** $\exists \theta \in \mathcal{M}$ *for which the following problems are $\mathcal{NP}$-hard:*

- *(P1): Determining the best global action:* $a_\theta^\star = \arg\max_{a \in \mathcal{A}} \sum_{e \in [\rho]} \theta_e(a_e)$.

- *(P2): Determining the $m$-best global actions, for $m \in [K^N]$.*

- *(P3): Given $\tilde{v} \in \mathbb{R}^{\tilde{A}}$, decide if $\tilde{v} \in \tilde{\mathcal{V}}$.*

*Proof.* (P1) can be reduced to the task of finding a Maximum A Posteriori (MAP) assignment in probabilistic graphical models, which is known to be $\mathcal{NP}$-hard [27, pag. 142]. Specifically, if $\theta \in \mathcal{V}$, i.e., it is an element of the $A - 1$ dimensional simplex, then (P1) corresponds to finding the MAP assignment for the measure $\theta$. If $\theta \notin \mathcal{W}$, we can find a mapping $f : \mathbb{R}^A \to \mathcal{V}$ s.t. $\theta' = f(\theta)$ and $a_\theta^\star = a_{\theta'}^\star$. Alternatively, (P1) can be cast as a Distributed Constraint Optimization Problem (DCOP), which is known to be $\mathcal{NP}$-hard [29]. Clearly, (P2) is $\mathcal{NP}$-hard since it can be reduced to (P1) for $m = 1$. Problem (P3) was shown to be $\mathcal{NP}$-complete by [34]. $\qquad \square$

## I An algorithm for selecting $\mathcal{A}_0$

We present in Alg. 4, the pseudocode of a simple procedure for selecting $\mathcal{A}_0$. It takes as input the set of global actions $\mathcal{A}$ and the set of group actions $\mathcal{A}_e$ for all $e \in [\rho]$. Let $I_{e,a_e} = \sum_{b \in \mathcal{A}_0} \mathbb{1}_{\{a_e = b_e\}}$ be the counter of group actions $a_e \in \mathcal{A}_e$ in $\mathcal{A}_0$, and define $I_e = (I_{e,a_e})_{a_e \in \mathcal{A}_e} \in \mathbb{N}^{A_e}$.

To describe the algorithm, we assume w.l.o.g. that $\mathcal{A}$ is an ordered set, and denote by $\mathcal{A}(i)$ the $i^{th}$ global action. First, the algorithm initializes $\mathcal{A}_0 \leftarrow \emptyset$, and $I_e = 0 \in \mathbb{N}^{A_e}, \forall e \in [\rho]$. Then, the algorithm iterates over groups $e \in [\rho]$ and groups' actions $b_e \in \mathcal{A}_e$, and iteratively includes arms in $\mathcal{A}$ into the set $\mathcal{A}_0$ which are never observed in previous iterates. By construction, Alg. 4, ensures that $\mathcal{A}_0$ contains global action covering every group action.

---
**Algorithm 4** BUILD $\mathcal{A}_0$

---
**Input:** Global actions $\mathcal{A}$, group actions $(\mathcal{A}_e)_{e \in [\rho]}$
**Initialize:** $\mathcal{A}_0 \leftarrow \emptyset$, $I_e = 0 \in \mathbb{N}^{A_e}, \forall e \in [\rho]$,
**for** $e \in [\rho]$ **do**
    $i \leftarrow 1$
    **while** $\min_{a_e \in \mathcal{A}_e} I_{e,a_e} = 0$ **do**
        $a \leftarrow \mathcal{A}(i)$
        **for** $b_e \in \mathcal{A}_e$ **do**
            **if** $b_e = a_e$ and $I_{e,b_e} = 0$ **then**
                $\mathcal{A}_0 \leftarrow \mathcal{A}_0 \cup \{a\}$
                $I_{e,b_e} \leftarrow I_{e,b_e} + 1$
        $i \leftarrow i + 1$
**Return** $\mathcal{A}_0$

---

Alg. 4 returns an exploration set that covers all group actions and that satisfies $|\tilde{A}_0| \leq \tilde{A}$. Note that Alg. 4 may be easily improved with more precise search strategies, at the cost of increased computational complexity. For example, the algorithm could include in $\mathcal{A}_0$, at each step, global arms $a \in \mathcal{A}$ which maximizes the number of non-observed group actions corresponding to the global actions in $\mathcal{A}_0$. When the set $\mathcal{A}$ is large, these refined searches may be computationally expensive.

*Example* 4 ($\mathcal{A}_0$ action choice). Consider the example in Fig. 8 with $K = 2$ local actions and $N = 4$ agents (i.e., $A = 16$ actions). In such setting, we can select $\mathcal{A}_0 = \{a_{0000}, a_{0110}, a_{1001}, a_{1111}\}$. Running Alg. 4 on this instance instead produces the set $\mathcal{A}_0 = \{a_{0000}, a_{0001}, a_{0110}, a_{0111}, a_{1000}, a_{1010}, a_{1100}\}$.

## J  MAMABs with specific graph structures

In this section, we present different MAMABs with specific graph structures.

### J.1  Acycic factor graphs

In this section, we introduce a few definitions and preliminary concepts on graphs from [39], and report examples of acyclic factor graphs. A *hypergraph* $G = (V, E)$ is a tuple containing: a vertex set $V = [N]$, and a set $E$ of hyperedges, where each hyperedge $e \in E$ is a particular subset of $V$. The *factor graph* associated to an *hypergraph* is a bipartite graph $G' = (V', E')$ with vertex set $V' = V \cup E$ and an edge set $E'$ that includes elements $(i, e)$, where $i \in V$ and $e \in E$, if and only if the hyperedge $e$ include vertex $i$. A *join tree* associated to a factor graph $G$, a.k.a. *junction tree*, is a tree $T = (V'', E'')$ such that the vertex set $V''$ corresponds to the factors of $G'$, and if two factors $e, e'$ include in their scope the same variable $i$ in $G'$, then every factor on the unique path between $e$ and $e'$ also include $i$ in $T$.

**Definition J.1** (Acyclic factor graph). A factor graph is said to be acyclic if it has a join tree.

Examples of acyclic factor graphs include the star and line factor graphs presented in Fig. 3, trees, and forest (i.e., ensembles of trees) depicted in Fig. 9.

### J.2  Networked Bandits

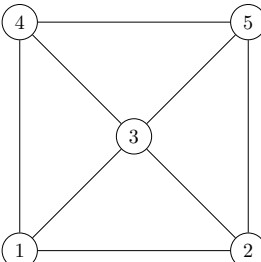

Figure 10: Example of reward graph in networked bandits.

We discuss a relevant particular case of MAMABs which we name Networked Bandits (NBs). NBs consist of a set of $N$ agents that are associated to an undirected graph $\mathcal{G} = (\mathcal{N}, \mathcal{E})$, where $\mathcal{N} = \{1, \ldots, N\}$ and $\mathcal{E} \subset \mathcal{N} \times \mathcal{N}$ is the set of edges. Each agent $i \in \mathcal{N}$, is associated with a local action $a_i \in \mathcal{A}_i$. We also denote the global action as $a = (a_1, \ldots, a_N) \in \mathcal{A}_1 \times \cdots \times \mathcal{A}_n$. For a node $i \in \mathcal{N}$, denote its neighbors by $\mathcal{N}_i$, and let $\mathcal{N}_{+i} = \mathcal{N}_i \cup \{i\}$. In our model, each agent is associated with a local reward function $r_i(a_{\mathcal{N}_{+i}})$ depending on the local action and on the actions of neighbors $a_{\mathcal{N}_{+i}}$. The reward experienced by node $i \in [N]$ is expressed as

$$r_i(a_{\mathcal{N}_{+i}}) = \theta_i(a_i) + \sum_{j \in \mathcal{N}_i} \theta_{i,j}(a_i, a_j) + \eta_i, \tag{32}$$

where $\eta_i \sim \mathcal{N}(0, 1)$, $\theta_i(a_i)$ is the average reward for the distribution of the $i$-th agent when she pulls action $a_i$, and $\theta_{i,j}(a_i, a_j)$ are the means of the distribution of the neighbor's terms influencing the reward by pulling action $a_j$. For a given set $\mathcal{S} \subseteq [N]$, we denote by $a_{\mathcal{S}} = (a_s)_{s \in \mathcal{S}}$. The NBs setting can be interpreted as a particular case of MAMABs in which $\rho = N$, and each group corresponds to a node and its neighbors.

### J.3 Reductions

The general MAMAB model studied in this paper encompasses the following particular cases:

(i) *Plain bandit:* For $\rho = 1$ and $\mathcal{S}_1 = [N]$, i.e., $N$ agents (each with $K$ local actions) connected to a single factor, the model reduces to a plain bandit model [25] with an exponentially large action space, i.e., $|\mathcal{A}| = K^N$.

(ii) *Uncoordinated bandit:* For $\rho = N$ and $\mathcal{S}_e = \{e\}$, for all $e \in [\rho]$, i.e., each group reward $r_e$ only depends on the action of a single agent $a_e \in \mathcal{A}_e : |\mathcal{A}_e| = K$.

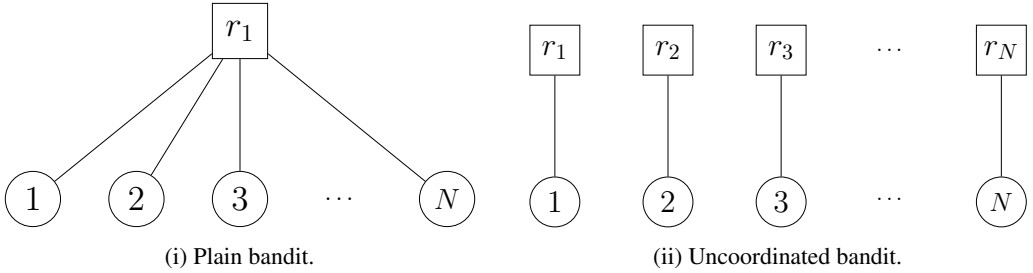

(i) Plain bandit.        (ii) Uncoordinated bandit.

Figure 11: Factor graphs for specific MAMAB models.

For (i), the global reward can be simply written as $\theta(a) = \theta_1(a_1, ..., a_N)$. Furthermore, the group variables and global variables coincide, i.e., $\tilde{\mathcal{V}} = \mathbb{R}_{\geq 0}^A$. The lower bound constant for the MAMAB model matches the one in the plain bandit setting [25], i.e., we have

$$C_\theta^\star = \inf_{v \in \mathbb{R}_{\geq 0}^A} \sum_{a \in \mathcal{A}} v_a \Delta(a) \quad \text{s.t.} \quad \frac{v_a^{-1}}{\Delta(a)^2} \leq 1, \forall a \in \mathcal{A}.$$

The uncoordinated setting (ii) corresponds to a game with $N$ independent bandits, each having observed rewards $\theta_i(a_i)$. Note that, as $\rho = N$, we can use the indices $e$ and $i$ interchangeably (each group contains a unique agent). We can write the global reward simply as $\theta(a) = \sum_{i \in [N]} \theta_i(a_i)$, and each agent can determine its best arm independently, i.e., an action $a \in \mathcal{A}$ is sub-optimal $a \neq a^\star \iff \exists i \in [N] : a_i \neq a_i^\star$. Then, the expression for the lower bound constant of (ii) can be written as $C_\theta^\star = \sum_{i \in [N]} C_{\theta_i}^\star$, where

$$C_{\theta_i}^\star = \min_{w_i \in \mathbb{R}_{\geq 0}^K} \sum_{a_i \in \mathcal{A}_i} w_{i,a_i}(\theta_i(a_i^\star) - \theta_i(a_i)) \quad \text{s.t.} \quad \frac{w_{i,a_i}^{-1}}{(\theta_i(a_i^\star) - \theta_i(a_i))^2}, \forall a_i \neq a_i^\star$$

Note that for $N = 1$ both (i) and (ii) reduce to a standard plain bandit model with $|\mathcal{A}| = K$.

## K Experimental settings and additional experiments

### K.1 Tightness of locally-tree like approximation

In this section we present experiments which highlight the tightness of the locally-tree like approximation $C_\theta^L$. We generate $N_{\text{sim}} = 100$ MAMABs instances, for a ring and a tree factor graph (see Fig. 3) with $N = 5$, $K \in \{3, 4\}$, when varying the parameters $\theta_e(a_e)$, for all $e \in [\rho], a_e \in \mathcal{A}_e$ uniformly at random in the interval $[0, 10]$, and comparing the lower bound and the locally tree-like approximation constants.

The results are presented in Fig. 12. As expected for the line factor graph the locally tree like approximation is tight, as this graph topology does not contain cycles. Note that, although the ring graph contains a cycle and hence $C_\theta^\star \leq C_\theta^L$, the locally tree like approximation is very close to $C_\theta^\star$ for most of the generated instances.

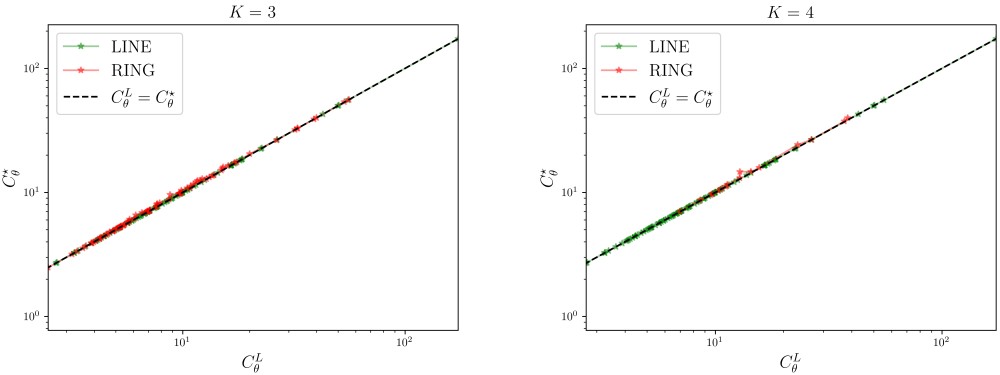

Figure 12: Comparison of $C_\theta^\star$ and $C_\theta^L$. Each point represent the lower bound constants for a MAMAB instance with a ring (green) or line (red) factor graph.

## K.2 Varying $m$ experiments

We report additional results on the regret of ESM for instances of MAMABs with a line and ring factor graphs with $N = 5$, $K = 3$, and varying $m$. As for the other experiments, the group parameters $\theta_e(a_e)$ are generated at random from the interval $[0, 10]$, for all $e \in [\rho], a_e \in \mathcal{A}_e$ and results are averaged over $N_{\text{sim}} = 5$ independent runs. The results are shown in Fig. 13. We use $\diamond = \text{MF}$ for the ring graph and $\diamond = \text{L}$ for the line graph.

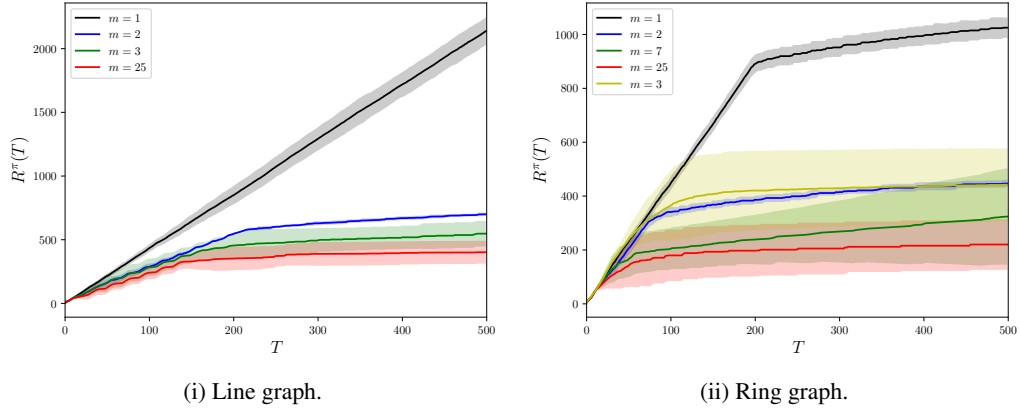

(i) Line graph.

(ii) Ring graph.

Figure 13: Experiments for varying $m$.

## K.3 Antenna tilt experiments simulation setting

The simulation settings used in the antenna tilt experiments are reported in Tab. 1.

Table 1: Simulator parameters.

| PARAMETER | SYMBOL | VALUE |
|---|---|---|
| Number of sectors | $|\mathcal{S}|$ | 6 |
| Number of UEs | $|\mathcal{U}|$ | 1000 |
| Antenna tilt values | $\mathcal{A}_i$ | $\{2°, 7°, 13°\}$ |
| Carrier frequency | $f$ | 1800 MHz |
| Antenna height | $h$ | 32 m |
| Network size | $M$ | 2 km$^2$ |

### K.4 Local approximation for cyclic factor graphs

In this section, we demonstrate experimentally that using $\diamond = L$ for ESM in MAMABs with cyclic factor graphs not attain a better regret than targeting exploration driven by $C_\theta^\star$, i.e. the true lower bound problem, even though $C_\theta^\star > C_\theta^L$ from the solutions of the lower bound optimization problems. We consider the cyclic factor graph in Fig. 14 (left) with $K = 3$ local actions. We compare the performance of ESM for and $\diamond = \{L, \diamond\}$. In Fig. 14 (center), we show the lower bound as $C_\theta^\diamond \log(T)$ and in Fig. 14 (right), we show the results for the regret of ESM when targeting $C_\theta^\star$ or $C_\theta^L$.

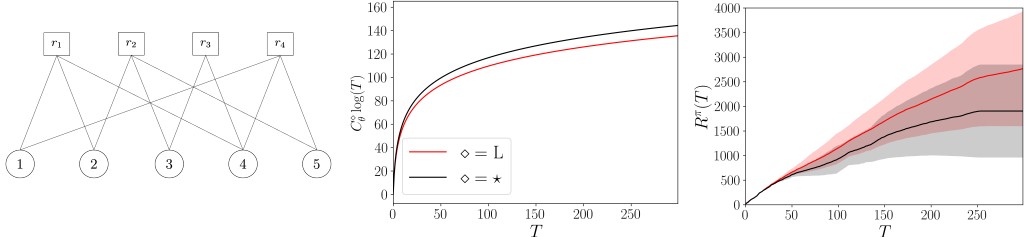

Figure 14: Left: cyclic factor graph considered for the experiments. Center: lower bound $C_\theta^\diamond \log(T)$, for $\diamond = \{L, \star\}$. Right: Performance of ESM for $\diamond = \{L, \star\}$.

In general, for many instances, we empirically observed that the values of $C_\theta^\star$ $C_\theta^L$ are very close. For example, in Fig. 12 (App. K), we compare these two quantities for 100 randomly generated instances (of the group means) for a ring factor graph (see Fig. 3).

### K.5 Quantifying the approximation ratio

In this section, we quantify the approximation ratio between the approximation constants $C^\diamond(m)$ for $\diamond = \{MF, L\}$, when varying $m$, and the true lower bound constant $C_\theta^\star$. We draw an instance of $\theta$ for a ring, line, and star graph (see Fig. 3) by selecting group means $\theta_e(a_e) \sim \mathcal{U}(0, M)$, for all $e \in [\rho], a_e \in \mathcal{A}_e$.

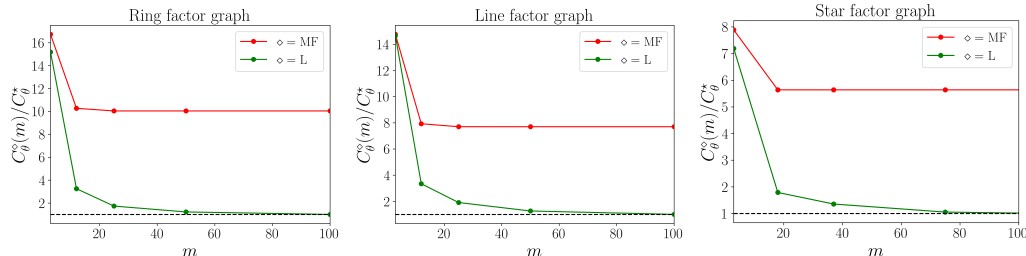

Figure 15: Approximation ratio $C_\theta^\diamond(m)/C_\theta^\star$ for $\diamond = \{MF, L\}$ and for different graph topologies ($K = 3$, $N = 5$). The dashed line represents $C_\theta^\diamond(m)/C_\theta^\star = 1$
.

The results presented in Fig. 15 show that $C_\theta^{MF}(m)$ is close to $C_\theta^\star$ (they are equal up to a small constant) and decreases with $m$. For $C_\theta^L(m)$, the same hold, and for $m$ large enough the approximation is tight, as predicted by our results.

# L  Details on antenna tilt optimization experiments

In this section, we present details on the antenna tilt optimization experiments.

**Throughput.** The throughput $T_{i,u}$, is formally defined in terms of the Signal-to-Interference-plus-Noise Ratio (SINR), a metric that measures the quality of a signal in the presence of interference and noise. Let $a_{\mathcal{N}_i}$ be the group vector containing

Specifically, the SINR of a UE $u \in \mathcal{U}$ connected to cell $c \in \mathcal{C}$ is defined as:

$$\mathrm{SINR}_{i,u}(a_i) = \frac{P_i G_{i,u}(a_i) L_{i,u}(a_i)}{\sum_{k \in \mathcal{N}_i} P_k G_{k,u}(a_k) L_{k,u}(a_k) + \sigma},$$

where $P_i$, $G_{i,u}$, and $L_{i,u}$ are the transmitter antenna power, the gain of the transmitter antenna, and path loss for UE $u$ connected to cell $i$, respectively. The gain is influenced by antenna parameters such as tilt and azimuth, and the path loss accounts for the transmission medium and obstacles (e.g., buildings, atmospherical conditions, vegetation, etc.). The throughput $T_{i,u}$ experienced by UE $u$ connected to the cell $i$ is then expressed as a function of the SINR and available bandwidth:

$$T_{i,u} = \omega_B n_{i,u}^R \log_2 \left(1 + \mathrm{SINR}_{i,u}\right),$$

where $n_{i,u}^R$ is the number of Physical Resource Blocks (PRBs) allocated to UE $u$ in cell $i$ and $\omega_B$ is the bandwidth per PRB (180 kHz). We use the average throughput of a cell in our group reward definition, i.e.,

$$r_i(a_e) = \frac{1}{|\mathcal{U}_i|} \sum_{u \in \mathcal{U}_i} T_{i,u}.$$

Hence the global reward is expressed as

$$r(a) = \sum_{i \in [N]} \frac{1}{|\mathcal{U}_i|} \sum_{u \in \mathcal{U}_i} T_{i,u}(a_e).$$

**On the noise independence assumption.** In our experiments, each group $e \in [\rho]$ corresponds to a sector: more precisely, it consists of an antenna $i \in [N]$ serving the users $u \in \mathcal{U}_i$ connected to this sector, and the set of antennas that can interfere with the transmissions of the antenna $i$.

Recall that the group reward is defined as $r_e(a_e) = \sum_{u \in \mathcal{U}_i} T_{i,u}(a_e)$, where $a_e$ represents the tilts of antennas in group $i$. The throughput $T_{i,u}(a_e)$ is the rate at which an user $u$ can decode transmissions from the antenna $u$. This rate depends on the random channel conditions (also known as *fading*) between each antenna in the group and the user $i$. Now the fadings between pairs of (antenna, user) are typically stochastically independent across users and antennas [32].

Since the sets of $(\mathcal{U}_i)_{i \in [N]}$ form a partition, they do not overlap, and the random variables $r_e(a_e)$ are indeed independent across groups. They can be modeled as independent Gaussian realizations in the sum-throughput over groups. For details, refer e.g., to [32].

**Additional details.** The set of UEs in the network is $\mathcal{U} = \cup_{i \in \mathcal{S}} \mathcal{U}_i$ as presented in Sec. 7.2. The number of UEs connected to cell $i$ is affected by tilt variation since we assume UEs connect to the cell from which they get maximum Reference Signal Received Power (RSRP). In particular, given a tilt configuration $a$, the UEs in cell $i$ are defined as

$$\mathcal{U}_i = \left\{ u \in \mathcal{U} : \arg\max_{k \in [N]} P_k G_{k,u} L_{k,u} = i \right\}.$$

There exist other methods to determine relations between antennas which rely on automated procedures, domain knowledge, and heuristics. For example, they may be based on the geographic distance between cells, on Neighbor Relations (ANR) as defined in 3GPP standards, on network planning tools for coverage prediction, or on cell handover logs [32]. In addition, domain knowledge can be used to refine the graph topology by pruning or adding edges based on key feature of a city or knowledge about the terrain (if there is a natural obstacle for example). Analyzing the influence of the graph structure is not in the scope of this paper and is left as future work.

## M Extended literature review

### M.1 MAMABs

As mentioned in Sec. 2, a few papers investigate MAMABs with the same factored reward structure as ours [2, 35, 37] for regret minimization.

Bargiacchi et al. [2] study a MAMABs setting in which the group rewards are random variables with finite support: $r_e(a_e) \in [0, r_{e,\max}]$, for all $e \in [\rho], a_e \in \mathcal{A}_e$. They propose MAUCE, an UCB-type algorithm, in which the bonus term is a non-linear sum of the group UCBs: at time $t$, MAUCE selects a global action

$$a_t = \arg\max_{a \in \mathcal{A}} \sum_{e \in [\rho]} \hat{\theta}_{t,e,a_e} + \sqrt{\frac{1}{2} \sum_{e \in [\rho]} \frac{r_{e,\max}}{N_{t,e,a_e}} \log(tA)}.$$

The non-linearity in the UCB term makes it difficult to use the VE to select the arm maximizing the UCB. They propose an efficient routine based on Multi-Objective Variable Elimination (MOVE) to compute the optimal UCB index. However, the computational complexity of the procedure is unclear. The asymptotic regret of MAUCE satisfies:

$$\liminf_{T \to \infty} \frac{R^{\pi_{\text{MAUCE}}}(T)}{\log(T)} \leq \frac{\Delta_{\min}^2 + 2\tilde{A}(\sum_{e \in [\rho]} r_{e,\max}^2)}{\Delta_{\min}^2}.$$

Assume that the means of the MAMABs parameters are bounded as $\theta_e(a_e) \in [0, r_{\max}], \forall e \in [\rho], a_e \in \mathcal{A}_e$, then our worst approximation ($C_\theta^{\text{MF}}(1)$), is better than a factor $r_{\max}$ with respect to the MAUCE bound. To see why, observe that by Lem. 5.4, we can upper bound $C_\theta^{\text{MF}}(1)$ as:

$$C_\theta^{\text{MF}}(1) \leq \rho \Delta_{\min}^{-2} \sum_{e \in [\rho], a_e \in \mathcal{A}_e} (\theta(a_e^\star) - \theta_e(a_e)) \leq 2\rho^2 \Delta_{\min}^{-2} \tilde{A} r_{\max},$$

while the leading constant of regret upper bound of MAUCE is $2\rho^2 \Delta_{\min}^{-2} \tilde{A} r_{\max}^2$.

Stranders et al. [35] study a similar setting with bounded rewards $r_e(a_e) \in [u_{\min}, u_{\max}]$, for all $e \in [\rho], a_e \in \mathcal{A}_e$. They propose HEIST, also an UCB-type algorithm. The asymptotic regret of HEIST satisfies

$$\liminf_{T \to \infty} \frac{R^{\pi_{\text{HEIST}}}(T)}{\log(T)} \leq \sum_{a \neq a^\star} \frac{8(u_{\max} - u_{\min})}{\Delta(a)},$$

which is clearly sub-optimal as it scales with the number of global arms $|\mathcal{A}|$.

Finally, [37] study the MAMAB setting in the Bayesian setting with sub-Gaussian rewards and propose Multi-Agent Thompson Sampling (MATS), a TS-based algorithm whose regret satisfies

$$R^{\pi_{\text{MATS}}}(T) \leq 2/\tilde{A} + \sqrt{64\sigma^2 \rho \tilde{A} T \log(\tilde{A} T)}.$$

Because of the differences in the problem formulations, this bound is not directly comparable to ours.

Finally, there are a few related works in MAMABs investigating the Best Arm Identification (BAI) problem [36, 3], where the objective is to identify the best global action with a prescribed error probability. In a closely related work [36], the authors derive a sample complexity lower bound defined through an optimization problem that is similar in structure to the one we derive for regret minimization and has exponentially many variables and constraints. Similar to our work, they derive an MF approximation of the lower bound problem. However, these MF approximations result in a non-convex optimization problem for the regret minimization setting (see App. B).

### M.2 Combinatorial Semi-Bandit Feedback

There is a large body of work [7, 10, 16, 40, 12, 13, 31, 38] investigating regret minimization in the (linear) combinatorial semi-bandit feedback. Although MAMAB is a more particular instance of bandits with combinatorial semi-bandit feedback (see App. E), the MAMAB combinatorial structure has been never considered in this setting to the best of our knowledge.

Most of these works focused on achieving a good trade-off between computational complexity and regret rates. None of these works focus explicitly on the MAMAB structure considered in our paper.

There exists algorithm using combinatorial versions of UCB. For example, the Combinatorial UCB (CUCB) [6] enjoys a $O(\tilde{A}\rho \log(T)/\Delta_{\min})$ regret guarantee. Combes et al. [10] propose the first (tight) lower bound in the combinatorial semi-bandit feedback setting and propose ESCB, an UCB-type algorithm which enjoys a $O(\tilde{A}\sqrt{\rho} \log(T)/\Delta_{\min})$ regret bound. Subsequently, Degenne et al. [16] tightens this regret bound to $O(\tilde{A}\log(\rho)^2 \log(T)/\Delta_{\min})$. All of the above-mentioned algorithms are sub-optimal and postulate the existence of a maximization oracle of the type

$$\max_{a \in \mathcal{A}} \sum_{e \in [\rho]} \theta_e(a_e) + \sqrt{\sum_{e \in [\rho]} \frac{c \log(T)}{N_{a_e}(t)}},$$

for some $c > 0$. The oracle is invoked at each time step, and hence, if such an efficient oracle exists (e.g., can be computed in polynomial time), the resulting algorithm is efficient. For many structures (e.g., $m$-sets, spanning trees, matroids, etc.) [13] showed that computing such maximum is an $\mathcal{NP}$-hard problem. As mentioned in App. M.1, for our MAMAB combinatorial structure, [2, 35] propose algorithms to compute such maximization, but without any computational complexity guarantee. Furthermore, Cuvelier et al. [13] proposes A-ESCB, which achieves a regret bound $O(\tilde{A}\log(\rho)^2 \log(T)/\Delta_{\min})$ by solving multiple times a budgeted linear optimization of the type $\max_{a \in \mathcal{A}: \sum_e c_e \mu_e(a_e) \geq s} \sum_{e \in [\rho]} \theta_e(a_e)$. In such case, the existence of an efficient oracle solving the budgeted linear maximization problem is assumed, which is true only for particular structures (specifically $s$-paths and $m$-sets).

Combinatorial bandit algorithms based on TS techniques [40, 31] usually require, at each time step, a maximization step of the type

$$\max_{a \in \mathcal{A}} \sum_{e \in [\rho]} \theta_e(a_e).$$

We show in Lem. H.1 that, for general factor graphs, performing this operation is $\mathcal{NP}$-hard.

Finally, Cuvelier et al. [12] derive the first asymptotically optimal algorithm for the semi-bandit feedback problem in polynomial time. We should mention that four our local approximation $C_\theta^\diamond$, [12, Assumption 6] is satisfied: $\exists M \in \mathbb{R}^{c \times d}$, $b \in \mathbb{R}^c$, with $c = O(\text{poly}(\tilde{A}))$: $\text{co}(\mathcal{A}) = \{\tilde{w} \in R^{\tilde{A}} : M\tilde{w} = b, \tilde{w} \geq 0\}$, i.e., the convex hull of the (global) action set can be represented by a polynomial number of inequalities. However, this approximation only holds for acyclic factor graphs, while the case of cyclic factor graphs is, to the best of our knowledge, not considered.