# OpenReview forum: "Statistical and Computational Trade-off in Multi-Agent Multi-Armed Bandits"
_NeurIPS.cc/2023/Conference — NeurIPS 2023 poster_

### Official Review · Reviewer_pqkD · 2023-07-06

**Soundness:** 4 excellent
**Presentation:** 4 excellent
**Contribution:** 3 good
**Rating:** 7
**Confidence:** 3

**Summary:**

This paper studies the Multi-Agent Multi-Armed Bandits problem with factor graph reward structure, which is motivated by the real-world antenna tilt optimization problem. It first proposes an asymptotic lower bound that involves an optimization problem with exponential number of variables and constraints. Then, this optimization problem is relaxed by approximation techniques motivated from probabilistic graphical model. Finally, this paper proposes an learning algorithm based on the approximated lower bound and supports it with both synthetic and real-world experiments.

**Strengths:**

- The relaxation contains novel application of various techniques from probabilistic graphical model to this concrete problem.
- The proposed algorithm can be computationally efficient and is flexible to lie at different points on the trade-off curve between statistical and computational complexity.
- The proposed algorithm enjoys good asymptotic regret and the theoretical guarantee is corroborated with real-world experiment.


**Weaknesses:**

Although I don't view this as a serious issue, it seems this paper focuses a lot in how to solve the problem in a computationally efficient way while the analysis of the algorithm's performance is a little bit limited. For example, we do not know how far the approximation is away from the exact lower bound and how the algorithm with locally tree-like reduction will perform in general.

### Suggestions on Writing
- It may be better to briefly mention the rough magnitude of $C^{\diamond}_{\theta}(m)$ again after stating Theorem 6.1.

**Questions:**

- What is the intuition behind using $\min_{a_i\in\mathcal{A}\_i}\frac{N_{t, i, a_i}}{w_{t, i, a_i}}$ as the action selection for exploration? In particular, how does this selection rule approximately satisfies the lower bound interpretation that action $a$ should expectedly be selected about $v^\star_a\log(T)$ times?
- It seems the varialbes $w=(w_i)\_{i\in[N]}$ is undefined for $C^{\diamond}_{\hat{\theta}_t}(m)$ when $\diamond=\mathrm{L}$. How will the ESM algorithm proceed when $\diamond=\mathrm{L}$?
- Is that possible to analyze how the algorithm will perform when $\diamond=\mathrm{L}$ and the factor graph is not an acyclic factor graph? (It certainly cannot be better than the lower bound.) Also, can we see how it will perform empirically?
- Is that possible to quantify the difference between $C^\star_{\theta}$ and $C^{\diamond}_{\theta}(m)$? That is, how far is the approximation away from the exact lower bound?
- In the antenna tilt experiment, does the choice of action set $\mathcal{A}_i=\lbrace 2^\circ, 7^\circ, 13^\circ\rbrace$ contain any special considerations (why these three degrees)?

**Limitations:**

The limitations are addressed well in this paper.

---

> ### Author Rebuttal · Authors · 2023-08-09
>
> We thank reviewer pqkD for the constructive and comprehensive review. We address questions in the order they are presented.
>
> - The intuition here is that in the exploration phase, we sample local actions that are the farthest from satisfying $N_{t,i,a_i} \approx w_{t,i,a_i} \log(T)$, for all agents $i\in[N]$. This is naturally achieved by selecting, local actions that minimize the ratio $N_{t,i,a_i}/w_{t,i,a_i}$. Now, the interpretation for the lower bound in terms of group variables $\tilde{v}$ (see Lem 4.2 and lines (124-126)) is that an optimal algorithm should sample each group arm $a_e$ for about $\tilde{v}^\star\_{e,a_e}\log(T)$ times. To see why the local action selection above will eventually satisfy this, we need to consider that (i) the condition in the exploitation phase of ESM enforces that $N_{t,e,a_e} \ge \tilde{v}\_{t,e,a_e}(1+\gamma)\log(T)$, and (ii) $C^\diamond\_{\hat{\theta}\_t}(m)$ is defined as an optimization problem over a constraint set $\tilde{\mathcal{V}}\_{\diamond}$ which imposes consistency between local variables $w$ and group variables $\tilde{v}$. These constraints are key to ensure that ultimately this local sampling will ensure that $N\_{t,e,a_e} \approx \tilde{v}\_{t,e,a_e}(1+\gamma) \log(T)$. We will include this discussion in the final version of the paper.
>
> - For $\diamond=L$, the variables $w=(w_{i,a_i})\_{i\in[N], a_i \in\mathcal{A}_i}$ are defined as $w\_{i,a_i} = \sum\_{b\in\mathcal{A}:b_e \sim a_i} \tilde{v}\_{e,b_e}, \forall e\in[\rho],\forall i\in\mathcal{S}_e,\forall a_i \in\mathcal{A}_i$ (recall that the notation $a_e\sim a_i$ means that the $i$-th element of $a_e$ equals $a_i$). This definition directly follows from the one of $\tilde{\mathcal{V}}\_L$ in Sec. 5.1.1.
>
>     In practice, when running ESM, we proceed as follows: to solve the optimization problem defined by $C^L\_{\hat{\theta}\_t}(m)$, we instantiate both the group variables $(\tilde{v}\_{t,e,a_e})_{e\in[\rho], a_e \in\mathcal{A}_e}$ and the local variables $(w\_{t,i,a_i})\_{i\in[N], a_i\in\mathcal{A}_i}$, and the corresponding constraints, as defined by $\tilde{\mathcal{V}}\_{L}$, so that both variables are naturally returned by the solver. We will include this clarification in the final version of the paper.
>
> - Thank you for your question. We acknowledge that quantifying the tightness of the approximation is an important point, which requires further investigation. However, we believe that addressing this point in its generality (for any $m$ and for both $\diamond = \\{L,MF\\}$) is very challenging due to the intricacy of the lower bound optimization problems involved in the approximations. Observe however that we presented different results on the quantification of our approximations in the paper, as summarized in the following points.
>
>     (i) We proved in Lem. 5.4 that for any $m$ and any $\diamond \in \\{L, MF\\}$, that $C^\diamond\_\theta(m+1)\le  C^\diamond\_\theta(m)$, i.e. the approximations tightens as we increase $m$, and that $C^\diamond\_\theta(K^N-1) = C^\diamond\_\theta$.
>
>     (ii) We proved in Lem. 5.1 that for acyclic factor graphs, it holds that $C^L_\theta = C^\star_\theta$ and hence the approximation is tight.
>
>     (iii) We proved in Lem. 5.3 the following scaling for the MF approximation: for any $\psi$ we have that $$C_\theta^{\psi-MF} \le \rho/\Delta_\min^2 \sum_{e\in [\rho], a_e\in\mathcal{A}_e}\theta_e(a_e^\star)- \theta_e(a_e).$$
>
>     We believe that these results complement each other and provide an overall understanding of the statistical properties of our approximations.
>
>     Finally, we would like to highlight that the main contribution of the paper is to break the combinatorial nature of the problem. In other words, going from a regret scaling as $\Theta(K^N)$ to one scaling as $\Theta(\rho K^d)$ (this scaling is guaranteed with our approximations) in a computationally efficient manner is the main contribution of the paper.
>
> - Thanks for the very interesting question. Analyzing the performance of ESM with $\diamond=L$ for factor graphs with cycles is actually one of the future directions we want to explore. Although we do not have any conclusive results on the analysis yet, we present a few empirical results in the attached document.
>
>     We consider the cyclic factor graph in Fig. 3 (left) of the document accompanying this rebuttal with $K=3$ local actions. We compare the performance of ESM for $\diamond=\\{\star,L\\}$ and $m=\tilde{A}$. In Fig. 3 (center), we show the lower bound as $C^\diamond_\theta\log(T)$ for $\diamond=\\{\star,L\\}$ and in Fig. 3 (right), we show the results for the regret of ESM. As expected, although $C^L\_\theta\le C^\star\_\theta$, $ESM(\diamond=L)$ cannot outperform $ESM(\diamond=\star)$.
>
>     In general, for many instances, we empirically observed that $C^{L}\_{\theta}$ and $C^\star\_\theta$ are usually close. For example, in Fig. 12 (App. K), we compare these two quantities for $100$ randomly generated instances (of the group means) for a ring factor graph (see Fig. 3). As shown in such figure, $C^L\_\theta$ is very close or indistinguishable from $C^{\star}\_{\theta}$ for all the instances. We observed similar behavior for other types of cyclic graphs. Based on these empirical observations, we believe that, potentially, it might be possible to use ESM with $\diamond=L$ even for cyclic factor graphs, and we will explore this direction in the future. We will include the above discussion in the final version of the paper.
>
> - The choice of the tilt angles in our experiments has no particularly deep explanation and is based on simulation-dependent observations.
> The choice of the maximum tilt ($13^\circ$) is also backed by the intuition that one may not want to select tilts with excessive values as these create blind spots in the coverage where network users are not connected to the network (see also App. L for additional details).  We will include this discussion in the final version of the paper.

---

> > ### Comment · Reviewer_pqkD · 2023-08-11
> > **Response**
> >
> > Thank you very much for your rebuttal and my concerns have been well-addressed!

---

> > > ### Author Response · Authors · 2023-08-11
> > >
> > > We are glad to have addressed your concerns. Thank you very much for taking our rebuttal into account.

---

### Official Review · Reviewer_EJNU · 2023-07-06

**Soundness:** 3 good
**Presentation:** 3 good
**Contribution:** 2 fair
**Rating:** 5
**Confidence:** 3

**Summary:**

This paper proposes the ESM algorithm for the regret minimization problem in Multi-Agent Multi-Armed Bandits, where the rewards are defined through a factor graph. The ESM algorithm attains a dedicate trade-off between statistical efficiency and computational efficiency in n Multi-Agent Multi-Armed Bandits.  The ESM algorithm is inspired by simple upper bounds of the regret lower bound, which characterizes the minimal expected number of times each global action should be explored and is computationally intractable to be exploited in the design of efficient algorithms. By tuning this upper bound, the ESM explores the trade-off between the achievable regret and the complexity of computing the corresponding exploration process. The ESM has a regret that asymptotically matches this upper bound. The regret and computational complexity of ESM are assessed numerically, using both synthetic and real-world experiments in radio communications networks.

**Strengths:**

This paper establishes a lower bound for the Multi-Agent Multi-Armed Bandits, where the rewards are defined through a factor graph.

Simple upper bound of the regret lower bound is established, which enables the design ESM.

Asymptotic regret upper bound of ESM is derived.


**Weaknesses:**

The regret lower bound looks not new and the proofs does not contribute new techniques. They look like straightforward applications of techniques from structured bandits.

Though upper bounds of the regret lower bound are established, the tightness of the upper bound is not proved. It is unknown how tight the upper bound is. Thus, it is unknown how much statistical efficiency is lost.

The regret upper bound of ESM is asymptotic, which looks much weak than the finite round regret upper bound.

The gap between the regret upper bound of ESM and the regret lower bound is not quantified. It is unknown how tight the regret upper bound is.



**Questions:**

Please refer to the weakness part.

**Limitations:**

See comments above.

---

> ### Author Rebuttal · Authors · 2023-08-09
>
> We thank reviewer EJNU for the comprehensive and detailed review. We address the main questions in the following.
>
> - *"The regret lower bound looks not new and the proofs does not contribute new techniques. They look like straightforward applications of techniques from structured bandits."*
>
> We agree that the lower bound is derived using classical techniques. We clearly state, after Theorem 4.1, that the bound is derived by using "classical change-of-measure arguments" and again in App. A, that the bound "is a direct consequence of an analogous bound in the combinatorial semi-bandit feedback setting, first given in [9, Th. 1] and leverages general techniques for controlled Markov chains".
>
> However, we do not understand why the reviewer is considering this point as a weakness. Presenting the lower bound serves two important purposes in the paper:
>
> 1. Previous works in the literature on MAMABs do not provide any lower bound.
>
> 2. More importantly, the lower bound is used in our paper as a starting point for the lower bound approximations and the algorithm design.
>
> - *"Though upper bounds of the regret lower bound are established, the tightness of the upper bound is not proved. It is unknown how tight the upper bound is. Thus, it is unknown how much statistical efficiency is lost -- The gap between the regret upper bound of ESM and the regret lower bound is not quantified. It is unknown how tight the regret upper bound is."*
>
> Thank you for raising this point. We are aware that a complete analytical quantification of the lower bound approximation is lacking and we believe that this is an important point for which further investigation is required. We would like to mention that addressing this point in its generality (for any $m$ and for both $\diamond \in \\{\text{L, MF}\\}$) is very challenging due to the intricacy of the lower-bound optimization problems involved in the approximations.
>
> However, we would also like to mention that we presented different results on the quantification of our approximations as summarized in the following points.
>
> (i) We proved in Lem. 5.4 that for any $m$ and any $\diamond \in \\{\text{L, MF}\\}$, that $C^\diamond_{\theta}(m+1)\le  C^\diamond_{\theta}(m)$, i.e. the approximations tightens as we increase $m$, and that $C^\diamond_{\theta}(K^N-1)\le  C^\diamond_{\theta}$.
>
> (ii) We proved in Lem. 5.1 that for acyclic factor graphs, it holds that $C^L_\theta = C^{\star}_{\theta}$ and hence the approximation is tight.
>
> (iii) We proved in Lem. 5.3 the following scaling for the MF approximation: for any $\psi$, we have that $$C^{\psi\text{-MF}} \le \rho/\Delta_{\min}^2 \sum_{e\in [\rho], a_e\in\mathcal{A}_e}\theta_e(a_e^\star)- \theta_e(a_e).$$
>
> We believe that these results complement each other and provide an overall understanding of the statistical properties of our approximations.
>
> In Fig. 2 of the accompanying document attached to this rebuttal, we show a sample of the approximation ratio $C^\diamond_{\theta}(m)/C^\star_{\theta}$ for $m \in \\{1, 3, 18, 37, 75, 100 \\}$, and $\diamond \in \\{\text{MF}, \text{L}\\}$ for different graph topologies. The results show that the $C^{\text{MF}}_{\theta}(m)$ is close to $C^{\star}\_\theta$ (they are equal up to a small constant) and decreases with $m$. For $C^\text{L}\_{\theta}(m)$, the same hold, and for $m$ large enough the approximation is tight $C^L\_{\theta}(m) = C^{\star}\_{\theta}$, as predicted by our results.
>
> Finally, we would like to mention that the main contribution of the paper is to break the combinatorial nature of the problem. In other words, going from a regret scaling as $\Theta(K^N)$ to one scaling as $\Theta(\rho K^d)$ (this scaling is guaranteed with our approximations) in a computationally efficient manner is the main contribution of the paper.
>
> - *"The regret upper bound of ESM is asymptotic, which looks much weak than the finite round regret upper bound"*
>
> Although we present the regret upper bound of ESM in the asymptotic form (as $T\to \infty$), we also obtain a finite time upper bound in App. D, as a byproduct of the proof of Thm. 6.1. The bound is: For any $\kappa >0$, and $T\ge 1$,
>
> $$
> R^{\pi}(T) \le \theta(a^\star) \left(M(\gamma,\rho) + \frac{2\tilde{A}}{\varepsilon\delta(\kappa)^2}\right) + ((1+\kappa)C^{\diamond}_{\theta} + 2 \varepsilon\psi^{\diamond}(\theta)(1+\gamma))\log(T),
> $$
>
> where $M(\gamma,\rho)$ is a constant depending only on the exploration constant $\gamma$ and the number of groups $\rho$, $\psi^{\diamond}(\theta)= \tilde{A} \\|\tilde{v}^\diamond(\theta) \\|\_\infty \sum_{e\in[\rho],a_e\in \mathcal{A}_e} (\theta_e(a^\star_e) - \theta_e(a_e))$ (see App. D for details). We will highlight the finite-time regret upper bound by using the extra space in the final version of the paper.

---

> > ### Comment · Reviewer_EJNU · 2023-08-15
> >
> > I do not buy your claim that "Previous works in the literature on MAMABs do not provide any lower bound."  There is a vast literature on MAMABs. It has been shown that the DPE algorithm is optimal for some variants of MAMABs, in the sense of of matching upper and lower bound[1].  Though they consider a setting different from this paper, it does not mean previous works on MAMABs do not provide any lower bound.
> >
> > To the best of my knowledge, MAMABs normaly study the setting that agents do not communicate or with restricted communication.  It seems that this paper does not consider such factors. To me, the setting of this paper is more like a variant of combinatorial bandits instead of MAMABs.
> >
> > This paper is placed more like a theoretical paper, since the experiments are just numerical simulations.  From a theoretical point of view, showing or even sufficient discussion on tightness of bound is of great value.  Furthermore, if the bound is not tight, it is difficult to judge how much statistical efficiency is lost and how insightful the result is.
> >
> > [1]  POAN WANG et al.  Optimal Algorithms for Multiplayer Multi-Armed Bandits, AISTAT 2020.

---

> > > ### Author Response · Authors · 2023-08-15
> > >
> > > Thank you for your comments.
> > >  - *I do not buy your claim that "Previous works in the literature on MAMABs do not provide any lower bound." There is a vast literature on MAMABs. It has been shown that the DPE algorithm is optimal for some variants of MAMABs, in the sense of of matching upper and lower bound[1]. Though they consider a setting different from this paper, it does not mean previous works on MAMABs do not provide any lower bound.*
> > >
> > > Regarding the lower bound. Apologies for the confusion in our rebuttal. We agree that for the MAMAB problem considered in [1] (that is very different than ours), lower bounds exist. What we intended to say in the rebuttal is that there exists no lower bound for the MAMAB problem considered in our paper, and also studied in [2, 3, 35]. What is important is that, as mentioned in our rebuttal, the lower bound is used in our paper as a starting point for the lower bound approximations and the algorithm design.
> > >
> > > - *To the best of my knowledge, MAMABs normaly study the setting that agents do not communicate or with restricted communication. It seems that this paper does not consider such factors. To me, the setting of this paper is more like a variant of combinatorial bandits instead of MAMABs.*
> > >
> > > The terminology “MAMAB” has indeed been used for different kinds of problems. In Wang et al. [1] (mentioned in your comments), all agents share the same action set and the reward distributions are the same across agents. The only interactions between the agents are when multiple agents select the same action (they experience a collision). That is why in [1], limiting the communication between agents is possible. The MAMAB we study is, from the agents’ coordination and learning perspective, more challenging because the rewards received by an agent truly depend on the actions selected by neighboring agents (i.e., agents in the same group). We agree that our MAMAB can be seen as a variant of a combinatorial bandit problem, as we explain at the beginning of the introduction. We also make a detailed connection between our problem and combinatorial bandits in Appendix E.
> > >
> > > - *This paper is placed more like a theoretical paper, since the experiments are just numerical simulations. From a theoretical point of view, showing or even sufficient discussion on tightness of bound is of great value. Furthermore, if the bound is not tight, it is difficult to judge how much statistical efficiency is lost and how insightful the result is.*
> > >
> > > Although the paper investigates the MAMABs problem theoretically, it also provide extensive simulations on the antenna tilt optimization problem in a realistic radio network simulator. We believe that these experiments are valuable to validate the practical applicability of our work. Note that this kind of experiments are not common in the bandit literature.
> > >
> > > About the tightness of the lower bound approximations, as we stated in the rebuttal, we believe that a complete characterization of the approximation is difficult. Nevertheless, we summarized our main results regarding the tightness of our approximations in the three points (i), (ii), and (iii) in the rebuttal.
> > >
> > > Finally, we would like to state that the main contribution of the paper is to break the combinatorial nature of the problem. Without exploiting the structure of the problem, one would get a regret scaling as $\Theta(K^N)$, hence exponentially growing with the number of agents. With our approach, we provably design a computationally efficient algorithm that achieves a regret scaling as $\Theta(\rho K^d)$.
> > >
> > > Thank you again for the discussion. We would be happy to answer further questions if any.

---

> > > > ### Comment · Reviewer_EJNU · 2023-08-16
> > > >
> > > > Thanks for the clarification. I will consider it in my final score.

---

> > > > > ### Author Response · Authors · 2023-08-18
> > > > >
> > > > > Many thanks for the discussion! Best wishes.

---

### Official Review · Reviewer_7bJY · 2023-07-06

**Soundness:** 3 good
**Presentation:** 3 good
**Contribution:** 3 good
**Rating:** 6
**Confidence:** 3

**Summary:**

This paper focuses on an interesting problem, which is a version of multi-armed bandits in which there are multiple agents and an action needs to be selected for each one of them. This is a problem that has been studied before in the literature and the authors here provide improved bounds on the regret by approximately a factor of k^d, where k is the number of agents and the degree of a graph that defines the reward function. Moreover, the authors' algorithm gives a knob that controls the tradeoff between the regret and the time spent solving the multi-agent optimization problem of each round.

The authors finally provide an experimental evaluation of their algorithm on a synthetic and a real dataset. The plots show that the regret is low and improves over a simple baseline and over previous work.

**Strengths:**

- The paper improves over the state of the art in theoretical guarantees
- The paper is well written, easy to follow
- The experimental evaluation shows good results

**Weaknesses:**

- The theoretical results are shown to be an improvement against [2] and [36] but the experimental part does not compare against [2] and [36].
- The related work pointers seem incomplete. The vast literature of combinatorial bandits seems to be very related (again combinatorial problems need to be solved in each round) but is not mentioned.

**Questions:**

- Why do you not compare against [2] and [36] in the experiments and rather choose weaker baselines?
- How does your problem compare against combinatorial bandits and why can't techniques from there be used in your setting?

---

> ### Author Rebuttal · Authors · 2023-08-08
>
> We thank reviewer 7bJY for the comprehensive and detailed review. We address the main questions in the following.
>
> - *"The theoretical results are shown to be an improvement against [2] and [36] but the experimental part does not compare against [2] and [36]. Why do you not compare against [2] and [36] in the experiments and rather choose weaker baselines?"*
>
> Thank you for noticing this issue, only due to a mistake we made when referencing a paper. Indeed, unfortunately, we realized that we misplaced a reference in the experiment section: [3] should be replaced by [2] in line 255. In our experiments, we do not choose a weaker baseline but the MAUCE algorithm shown in Figs. 4,5,6, actually corresponds to the reference [2]. We thank the reviewer for highlighting this issue, which we will fix in the final version of the paper.
>
> Now, to the best of our knowledge, the MAUCE algorithm [2] is the state of the art in MAMABs and the best competitor to our algorithm. As stated in lines (73-75), the HEIST algorithm proposed in [36] is very similar to MAUCE: it is a UCB-type algorithm but with a different constant in the UCB term. However, HEIST has worse regret guarantees for most MAMABs instances: its asymptotic regret scales as $O(K^N\Delta_{\max}/\Delta_{\min}\log(T))$, while the scaling of the regret of MAUCE is $O(\rho^2 K^d\Delta_{\max}^2/\Delta_{\min}^2\log(T))$ (please refer to Sec. 2 and App. M.1 for details). Hence, in the experiments, we choose to compare our algorithm to MAUCE only. However, for the sake of completeness, we have now performed a few additional experiments testing the performance of HEIST [36]. The results of these experiments are included in Fig. 1 of the accompanying document attached to this rebuttal. We will include these experiments in the final version of the paper.
>
> - *"The related work pointers seem incomplete. The vast literature of combinatorial bandits seems to be very related (again combinatorial problems need to be solved in each round) but is not mentioned. How does your problem compare against combinatorial bandits?"*
>
> Thank you for your question. Although we acknowledge your concern, we believe that related works in combinatorial bandits (and the relationship between combinatorial bandits and MAMABs) have been covered in the paper and the appendices, as we explain below.
>
> 1. First, the connection between combinatorial bandits and MAMABs is presented in App. E, where we explain that our MAMAB setting can be interpreted as a particular instance of combinatorial bandits with semi-bandit feedback, and we provide an example of this connection.
> 2. Second, in lines (78-87) of the related work section, we mention different related works in the combinatorial semi-bandit feedback literature ([24, 9, 11, 12, 35]) and we discuss the closest and most relevant work [11] in the main body of the paper. In the current submission, other related works in combinatorial bandits are discussed in detail in App. M.2, for reasons of space.
>
> We will include this discussion in the main body of the paper by using the extra space in the final version of the paper. We have further included a few additional references in the answer to the next question, which are reported at the end of this rebuttal. Moreover, should the reviewer be aware of other relevant works in combinatorial bandits that are not already covered in the current manuscript, we would welcome the opportunity to include them.
>
> - *"Why can't techniques from there be used in your setting?"*
>
> We believe that this point has been addressed in App. M.2 and App. H, and we are glad to provide further clarification. As the reviewer suggests, in principle, one could indeed use combinatorial bandit methods as those referred into Sec. 2 or App. M.2, but most likely these will yield computationally inefficient algorithms, as we explain in the following.
>
> For example, combinatorial bandit algorithms based on Thompson Sampling (TS) techniques [34, a, b] usually require, at each time step, a maximization step of the type $\max_{a\in\mathcal{A}} \sum_{e\in[\rho]} \theta_e(a_e)$. We show in Lem. H.1 that, for general factor graphs, performing this operation is $\mathcal{NP}$-hard.
>
> Other combinatorial bandit algorithms, employing e.g., Upper Confidence Bound (UCB) methods such as [16, 9, c], often involve even harder maximization steps w.r.t. the maximization presented above. For example, the algorithms proposed in [16, 9, c] require solving an index maximization problem of the type
>     $$
>     \max_{a\in\mathcal{A}}  \sum_{e\in[\rho]} \theta_e(a_e) + \sqrt{\sum_{e\in[\rho]} c\log(T)/N_{e,a_e}(t)}.
>     $$
> For many combinatorial structures (e.g. even for $m$-sets), there is no polynomial time algorithm to solve the task above [11]. We will include the above discussion in the final version of the paper.
>
> **Additional references**
>
> [a] Siwei Wang and Wei Chen, *Thompson sampling for combinatorial semi-bandits*. In Proc. of ICML, 2018.
>
> [b] Pierre Perrault,  Etienne Boursier, Vianney Perchet, Michal Valko, *Statistical Efficiency of Thompson Sampling for Combinatorial Semi-Bandits*. In Proc. of NeurIPS, 2020.
>
> [c] Wei Chen, Yajun Wang, Yang Yuan. *Combinatorial Multi-Armed Bandit: General Framework, Results and Applications*. In Proc. of ICML, 2013.

---

> > ### Comment · Reviewer_7bJY · 2023-08-11
> >
> > Thank you for your response, you indeed answered the raised questions. I will revise my score accordingly.

---

> > > ### Author Response · Authors · 2023-08-11
> > >
> > > Many thanks for considering our rebuttal and for  revising the score.

---

### Official Review · Reviewer_Pi4y · 2023-07-07

**Soundness:** 3 good
**Presentation:** 3 good
**Contribution:** 3 good
**Rating:** 6
**Confidence:** 2

**Summary:**

This paper consider the regret minimization problem in MAMABs, which typically has an exponentially large action space in relation to the number of agents. The authors first establish a lower bound result for general MAMAB by generalizing techniques from single agent bandit literature. More precisely, the lower bound is obtained by solving a convex optimization problem with exponentially many variables and constraints. Next, the authors present an equivalent characterization of the program for reward structures with acyclic factor graphs and provide mean-field approximation programs for general graph models. Finally, the authors design computationally tractable algorithms based on the proposed equivalent(approximation) program for acyclic factor graphs(general graphs).

**Strengths:**

1. Most writing is clear and easy to follow, even for those without extensive knowledge in MAMABs. Most claims and definitions are well-supported by existing literature.

2. Both the proposed lower bound and the proposed approximation scheme are interesting.

3. Authors also provide simulations and real-world applications to further enhances the relevance and practicality of the proposed methods.

**Weaknesses:**

While the paper overall is well-written and informative, I found some parts of the presentation to be unclear. Specifically, there were several points where I had questions and would have appreciated more explanation or clarification.


**Questions:**

1. What is the trade-off when letting $\varepsilon, \gamma\to 0$ in Algorithm 1? It appears that Theorem 6.1 always encourages selecting small values for $\varepsilon,\gamma$

2. In the acyclic factor graphs setting, while the proposed characterization is statistically tight, what is the computational complexity of the proposed algorithm to achieve optimal regret(i.e. select $m = K^N - 1$)?  Will it achieve the same polynomial-time complexity as in [11] in this setting?

3. In line 155, authors said "Unfortunately, for general graphs, we have that $C^L_\theta < C_\theta^\star$ (a direct consequence
156 of [36, Prop. 4.1]), and hence it is impossible to devise algorithms based on this approximation."
I am wondering why it is impossible to propose an approximation algorithm based on the  $C^L_\theta$ value (since the proposed approximation program in the mean-field approximation is also not guaranteed to have a value equal to $C_\theta^\star$)


**Limitations:**

Yes

---

> ### Author Rebuttal · Authors · 2023-08-08
>
> We thank reviewer Pi4y for the constructive and comprehensive review. We clarify the main questions in the following.
>
>
> 1. *"What is the trade-off when letting $\varepsilon,\gamma \to 0$ in Algorithm 1? It appears that Theorem 6.1 always encourages selecting small values for $\varepsilon,\gamma$*
>
> The choice of $\varepsilon, \gamma$ in our algorithm does not really involve any trade-off. These parameters are "safety parameters" that should just be strictly positive for our analysis to hold. The parameters impact the amount of exploration performed by ESM. When decreasing both these parameters the exploration of ESM also decreases, and this in turn improves the statistical performance of the algorithm. More precisely, in the proof of Thm. 6.1 (see App. D), we show that the regret upper bound is minimized when $(\varepsilon,\gamma) \to (0,0)$. We will include these clarifications in the final version of the paper.
>
> 2. *"In the acyclic factor graphs setting, while the proposed characterization is statistically tight, what is the computational complexity of the proposed algorithm to achieve optimal regret (i.e. select $m = K^N-1$)? Will it achieve the same polynomial-time complexity as in [11] in this setting?*"
>
> In the acyclic factor graph setting, the leading term in the computational complexity of ESM is given by the cost of determining the $m$ smallest gaps. Finding the best algorithm for solving this task is still an active area of research (see [15] and references therein for details). For example, the elim-$m$-opt algorithm has complexity $O((m + 1)NK^{A_{\mathcal{O}}+1})$ (see line 203 and App G). Hence, for $m = K^N-1$ the computational complexity of ESM in this case would be exponential in $N$. In fact, in practice, our method is meant to be applied when $m$ does not grow exponentially in $N$ (we will clarify this), because if it does, then the algorithm we obtain is at least as complex as the initial lower bound problem. As explained in Sec. 5.2, in practice selecting $m \simeq \tilde{A}$ yields a good trade-off between statistical complexity and computational complexity (see Fig. 2), and the latter will be polynomial in this case.
>
> About the computational complexity of [11], we believe that even for acyclic factor graphs, the algorithm of [11] would not lead to a polynomial-time computational complexity. Indeed, to prove a polynomial sample complexity of their algorithm, the authors postulate the existence of a *"budgeted linear maximization"* oracle (Assumption 2 in [11]), and we are not aware of the existence of such an oracle for these types of MAMABs instances. We will include these considerations in the final version of the paper.
>
> 3. *"In line 155, authors said "Unfortunately, for general graphs, we have that $C^L_{\theta} < C^\star_{\theta}$ (a direct consequence 156 of [36, Prop. 4.1]), and hence it is impossible to devise algorithms based on this approximation." I am wondering why it is impossible to propose an approximation algorithm based on the $C^L_{\theta}$ value (since the proposed approximation program in the mean-field approximation is also not guaranteed to have a value equal to $C^{\star}_{\theta}$)."*
>
> This is an interesting question! We cannot devise algorithms based on $C^{\text{L}}_{\theta}$ for MAMABs instances with cyclic factor graphs because it would contradict the lower bound result in Thm. 4.1.
>
> To see why, note that:
>
> (i) for acyclic factor graphs, we have that $C^L_{\theta} < C^\star_{\theta}$, and
>
> (ii) the lower bound dictates that any (consistent) algorithm must satisfy $R^{\pi}(T)/\log(T) \ge C^\star_{\theta}$ as $T\to \infty$ (see Thm. 4.1).
>
> Now, if one could devise an algorithm based on ESM which target $C^L_{\theta}$, we would get an algorithm with an upper bound than that attains $R^{\pi}(T)/\log(T) \le C^L_{\theta} < C^\star_{\theta}$ as $T\to \infty$ (see Thm. 6.1), which clearly contradicts the lower bound.
>
> On the other hand, for the Mean-Field (MF) approximation, we have that $C_{\theta}^{\psi\text{-MF}} >  C^\star_{\theta}$, for any $\psi$, and hence the problem described above for $C^L_{\theta}$ is avoided. We will include these clarifications in the final version of the paper.

---

> > ### Comment · Reviewer_Pi4y · 2023-08-17
> >
> >  I would like to thank the authors for the helpful response. My original score remains unchanged.

---

### Author Rebuttal · Authors · 2023-08-09

We thank the reviewers for their comprehensive and detailed reviews. We address the reviewers' questions individually in the rebuttals below. As requested by the reviewers, we have run additional experiments during the rebuttal period. We present the results of these experiments in the accompanying document attached to this rebuttal.

---

### Decision · Program_Chairs · 2023-09-21

**Decision:**

Accept (poster)

**Comment:**

The reviewers agreed in the opinion that the paper makes good progress on the multi-agent bandit problems and gives an interesting insight on the tradeoff between the regret and the computational efficiency. On the other hand, some concerns on the contribution of the paper in relation with existing work are raised. They seem to be largely solved by the rebuttal but I expect that the authors carefully incorporate them in the final version.